# Mechanical response characteristics of full depth reclaimed pavement based on 3D-move model

Haiwei Zhang[1,2,3]*, Ning Liu[1], Bowei Sun[4], Qingqing Zhang[1], Jiazhen Liu[1]

1 School of Civil and Environmental Engineering, Zhengzhou University of Aeronautics, Zhengzhou, Henan, China, 2 Henan Provincial Engineering Laboratory of High Permeability Pavement Materials, Zhengzhou, Henan, China, 3 Henan Provincial Engineering Technology Research Center of Modified Asphalt Pavement Materials, Zhengzhou, Henan, China, 4 School of Transportation Science and Engineering, Civil Aviation University of China, Tianjin, China

* zhanghw@zua.edu.cn

## Abstract

The mechanical properties of full-depth reclamation of pavements with Portland cement (FDR-PC) provide a theoretical basis for seeking the structural design of reclaimed pavements and selecting specific technical solutions. Relying on the typical reclaimed pavement structure, this paper applies 3D-Move Analysis to construct a numerical model of full-depth reclaimed pavement. It dynamically analyzes the mechanical properties of FDR-PC in four aspects: plane, spatial, speed, and temperature. The study shows that: in the plane direction, only the mechanical response at the center of the wheel track can be investigated; the maximum vertical strain along the depth direction from the road surface to the top of the residual layer occurs at the top of the residual layer of the old road; the speed is within a certain range, and the faster the vehicle travels, the smaller the influence on the mechanical response; when the temperature increases, the individual mechanical response increases to different degrees.

## 1 Introduction

Cold reclamation is a key technology for repairing, maintaining, and preventing the increase of deteriorated asphalt pavements. The main advantages of this technology are that it can reduce economic costs and fuel emissions and improve pavement performance [1]. Cold reclamation can reduce the environmental impact by about 15–18% in terms of acidification and abiotic consumption of fossil fuels and energy consumption [2]. The cold reclamation process reuses materials from old highways, less material is transported, and the reduction in heavy vehicle traffic produces less noise and greenhouse gas emissions [3,4].

Distinguished from cold reclamation, full-depth reclamation (FDR) focuses more on subgrade destruction. Full depth reclamation technique mixes the asphalt surface with a portion of the subgrade, so that the reclaimed material is not only mixed with

**Data availability statement:** The datasets generated and analyzed during the current study are available in the Zenodo repository (URL: https://zenodo.org/records/15022657).

**Funding:** This research was funded by the Science and Technology Department of Henan Province (NO.242102241013) and Henan Provincial Department of Housing and Ur-ban-Rural Development Science and Technology Planning Project (K-2359),and Supported by the Youth Research Funds Plan of Zhengzhou University of Aeronautics (23ZHQN01008).

**Competing interests:** The authors have declared that no competing interests exist.

the reclaimed asphalt material, but also with the reclaimed subgrade material [5]. Full-depth reclamation of pavements with Portland cement (FDR-PC) is a cold treatment technique that can treat most of the distresses in old asphalt pavements [6].

Full-depth reclamation of pavements with Portland cement uses special equipment to loosen the asphalt layer and part of the lower bearing layer in situ, or part of the asphalt layer is partially or completely milled to remove part of the lower bearing layer in situ loosening, and at the same time mixed with a certain amount of new mineral, recycled binding material, water, etc., after the mixing, paving, compaction and other processes at room temperature. The technology of regeneration of old asphalt pavement is realized through the processes of mixing, paving, and compaction at room temperature [7]. This is a cold regeneration technology that can deal with most of the old asphalt pavement diseases. The constituent materials of the new base are reclaimed asphalt pavement (RAP), recycled base material, binder material (emulsified asphalt, foam asphalt, or cement), and water (optimum content for compaction). The amount of RAP in the mixture depends on the thickness of the asphalt layer. Compared to other regeneration techniques it has the advantages of low production cost, high operational efficiency, low traffic disturbance, and the ability to improve the shape of the pavement while repairing it, focusing on the integrated recycling of semi-rigid base material and asphalt material [6].

Mallick RB et al [8] conducted indirect tensile strength tests under dry and wet conditions on specimens of FDR mixed with water, emulsion, emulsion and lime, cement and emulsion, and cement, and the results showed that FDR-PC had the highest tensile strength ratio. Jones D et al [9] by comparing the accelerated loading test study of FDR-PC and full-depth regeneration technology without stabilizers, the results showed that the rutting performance and stiffness are superior to full-depth regeneration technology without stabilizers.

Ghanizadeh AR et al [10] investigated the effect of different percentages of cement and recycled asphalt pavement on the optimum moisture content, maximum dry density, and unconfined compressive strength (UCS) of FDR-treated subgrade through a study using two different types of agglomerated soils. The results showed that increasing the amount of cement both led to a decrease in the optimum moisture content and maximum dry density of the treated subgrade, and the 7-day and 28-day UCS increased with increasing the amount of cement for a certain amount of RAP dosing, and the higher the proportion of RAP material in the FDR layer, the higher the proportion of silicate cement needed, and increasing the cement or decreasing the amount of RAP content increased the modulus of elasticity of the treated subgrade. Fedrigo W et al [11] compared the flexural strength, strain at break, and flexural modulus of cement-treated RAP and lateritic soil (LS) mixtures, and showed that the flexural strength and resilient modulus increased with the increase in the RAP dosage; the strain at break increased with the increase in RAP and the decrease in the cement dosage, increasing the mixture's flexibility; the higher the cement content, the greater the strength and stiffness of the mix. The strain-based fatigue relationship was obtained. Mechanistic analysis showed that the fatigue life of RAP and LS cement stabilized recycled base increased with the

increase of RAP dosage, asphalt wearing course thickness and recycled base thickness. The effect of cement content depends on the thickness of the layer. Euch Khay SE et al [12] formulated different cement-treated RAP mixes using local sources, and then these mixes were subjected to conventional mechanical tests (compressive strength, flexural strength, indirect tensile strength, and modulus of elasticity), and the results showed that an increase in RAP dosing led to a decrease in compressive strength, flexural strength, and indirect tensile strength. López MAC et al [13] conducted static and cyclic flexural loading tests on six different mixes with different cement dosages and RAP admixtures, and the results showed that the fatigue life of cement stabilized recycled subgrade depended on the composition of the mix, the thickness of the asphalt wearing course and the recycled layer. A study by Eugene A. Amarh EA et al [14] showed that FDR-PC not only improves strength but also reduces the effect of temperature on FDR modulus.

There are few research results on the mechanical properties of full-depth reclaimed pavement structures, and there is no detailed discussion. In China, there is no mature design method for FDR-PC, and they are all designed according to the reconstructed pavement, which restricts the application and development of the full-depth cold regeneration technology to a certain extent.

Studying the mechanical properties of FDR-PC pavements can understand the behavior of reclaimed pavements when subjected to traffic loads. This is essential for designing and planning road projects to ensure that the reclaimed pavement will meet the structural strength and stability requirements in actual use. Imad L. Papagiannakis A et al [15] showed that the application of finite element techniques can take into account almost all control parameters and can more accurately simulate complex material properties and realistic tire loads. The 3D-Move program [16] developed by the University of Nevada, Reno is well suited to analyze the mechanical response of pavements compared to more sophisticated and general-purpose finite element programs such as ABAQUS. Hajj EY et al [17] applied 3D-Move to compare the response of asphalt pavement at three different downhill speeds (64km/h, 32km/h, and 3.2km/h) of trucks. Siddharthan RV et al [18–21] presented many important pavement response parameters generated by 3D-Move under various loading conditions, verified the applicability of the chosen finite layer mechanics model used to perform pavement response calculations through 3D-Move, and established a traffic-induced pavement Strain database. Nasimifar M et al [22] used 3D-Move program to simulate TSD loading by taking into account the moving load, the uneven contact pressure of the dual tires and the viscoelastic material properties of the asphalt layer. Ghanizadeh AR et al [23] performed a quasi-static analysis of the flexible pavement by using the frequencies obtained from the FFT method and compared it to the fully dynamic analysis performed using 3D-Move program for full dynamic analysis. Kuchiishi AK et al [24] input the dynamic modulus results into 3D-Move Analysis program to study the pavement mechanical behavior of cold recycled asphalt mixtures. Zihan ZU et al [25] studied the pavement mechanical behavior of asphalt mixtures by measuring the dynamic modulus of asphalt mixtures and inputting it into 3D-Move Analysis program for calculation, verified the feasibility of converting TSD deflection to FWD degree. In 2014, Bai L [26] used 3D-Move Analysis software to establish a three-dimensional pavement model on the basis of vehicle dynamic loading test, and applied dynamic loads on the simulated pavement to investigate the actual behavioral characteristics of asphalt pavement structure under dynamic loading conditions.In 2017, Yan KZ [27] and others applied 3D-Move Analysis program to establish viscoelastic material properties under the Mechanical response model, compared and analyzed the effects of different load contact forms on the pavement structure shear stress, road surface bending settlement, tensile stress at the bottom of the surface layer, and the maximum value and location of compressive strain at the top of the soil base under static and dynamic loading.In 2019, Huang ZY et al [28] conducted a mechanical response analysis of thermally regenerated asphalt pavements under dynamic loading by using 3D-Move Analysis finite layer program.

The purpose of this thesis is to carry out simulation and analysis through 3D-Move Analysis program, establish calculation model relying on typical reclaimed asphalt pavement structure, analyze and compare the performance of FDR-PC, and calculate the time-course information of the mechanical response of the surface vertical displacement of pavement structure, and the stress and strain of the bottom of each layer of the pavement structure according to the model

constructed, in order to seek for the key mechanical indexes of the pavement and the most unfavorable state of the force, and to analyze the spatial force characteristic of the pavement, so as to provide theoretical basis for the design of the structure of reclaimed pavements and selecting the specific technological solutions.

## 2 Description of the 3D-Move program

3D-Move program is a professional software developed by the University of Nevada to study the mechanical analysis of asphalt pavement structures, which can consider some important factors affecting the mechanical response of pavement structures. Such as moving loads, three-dimensional contact stress distributions (both normal and shear stresses) of arbitrary shapes, and pavement viscoelastic material properties.

The continuum-based finite layer method uses the Fourier transform technique as the analysis method. The finite layer method is more computationally efficient than the moving load model based on the finite element method. This is due to the fact that pavements are usually horizontally layered and the pavement response is usually required at only a few selected locations; for such problems, the finite layer method of three-dimensional movement analysis is very suitable. Since the method can be applied to viscoelastic materials, it is an ideal tool for modeling the performance of asphalt concrete layers as well as for studying the variation of pavement response with vehicle speed. The 3D-Move Analysis employs a frequency-domain solution that allows the test data of asphalt mixtures to be used directly in the analysis.

The 3D-Move Analysis program describes the pavement layer system as consisting of multiple viscoelastic or elastic horizontal layers, each described using uniform properties. Each layer has a constant thickness and extends horizontally to infinity. The software uses a left-handed Cartesian coordinate system with the X-axis (longitudinal) parallel to the direction of travel, the Y-axis (horizontal) perpendicular to the direction of travel, and the Z-axis positive in the downward direction. Layers are numbered from top to bottom with no gaps in between, assuming perfect bonding between layers. The top surface is set to zero in the Z-axis direction, which follows traditional geotechnical sign conventions. Compressive stresses and strains are positive, while tensile stresses and strains are negative.

The primary input parameters to the 3D-Move program include analysis type (static or dynamic), loading configuration (total load, tire contact shape, and vehicle speed), and material properties (thickness and modulus).

## 3 3D-Move simulation

### 3.1 Establishment of pavement structure analysis model

With reference to the typical semi-rigid base layer asphalt pavement structure in China, this paper adopts a thin surface layer scheme, aiming to increase the influence of the reclaimed base layer on the structural performance and to highlight the possible effects of small changes in the mechanical behavior of the reclaimed mixture. The pavement structure is designated as the surface layer, the recycled base layer, and the old road residual layer. The surface layer was divided into the upper(4 cm) and lower(6 cm) layers. The old road residual layer consists of the old road subgrade and the old road base that have not been reclaimed and utilized. These two together form the subgrade of the new road (Table 1).

This paper focuses on simulation calculation using simulation software, and the relevant parameters are not directly obtained through experiments, but indirectly obtained by relying on references or previous experience. The pavement structure is shown in Fig 1.

### 3.2 Load shape and magnitude settings

The concept of cumulative equivalent axle loads is used in pavement design. Highway vehicles traveling on a variety of types, different models and different role of the number of different impacts on the road surface, in order to facilitate the design of the road surface, the need for a combination of different models of mixed traffic converted to some kind of uniform axle load equivalent axle times.

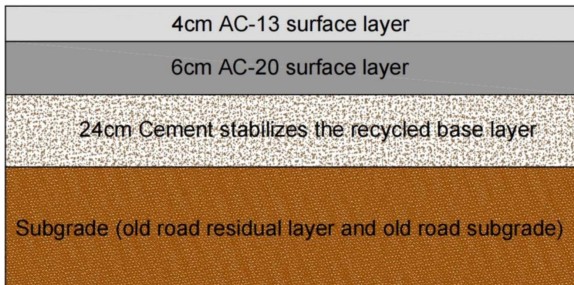

**Fig 1. Schematic of pavement structure.**

**Table 1. Pavement structure design.**

| Layer No | Layer Type | Material | Thickness (m) |
|---|---|---|---|
| 1 | Asphalt | Linear Elastic/Viscoelastic | 0.04 |
| 2 | Asphalt | Linear Elastic/Viscoelastic | 0.06 |
| 3 | Base | Linear Elastic | 0.24 |
| 4 | Subgrade | Linear Elastic | 0 (Semi-infinite) |

In this paper, the configuration of BZZ-100 single-axle-double-wheel set, first of all, the axle load, tire pressure, equivalent circle radius settings, the equivalent circle radius is 0.107 m, the tire ground pressure 700 kPa, the standard axle load of 100 kN, allocated to each tire for 25 kN.

### 3.3 Analysis type

In this study, the default moving speed is set to 60km/h for dynamic analysis as a way to investigate the mechanical properties of recycled asphalt pavement.

### 3.4 Material properties

**3.4.1 Surface parameters.** The performance of pavement materials is a key factor in analyzing the mechanical response of pavements. It is known that the surface layer will be a two-layer structure, divided into a 4 cm upper layer and a 6 cm lower layer. The surface layer uses asphalt mixture as the material, and asphalt is a viscoelastic material, the modulus of this material is affected by the temperature and the frequency of loading. The modulus of asphalt is different at different temperatures and loading frequencies. At high temperatures, the modulus of asphalt is close to that of granular materials, and at low temperatures, the modulus of asphalt is close to that of concrete. The modulus of asphalt material becomes larger with the increase of loading frequency. Therefore, for viscoelastic materials, the dynamic modulus is used to replace the compressive rebound modulus to respond to its mechanical properties more in line with the engineering reality.

Dynamic modulus test is used to characterize the surface asphalt mixture, according to the experimental results of our subject group, the reference temperature is set to 20°C, and four temperatures were selected as 5°C, 25°C, 35°C, 50°C and six different loading frequency (0.1 Hz, 0.5 Hz, 1 Hz, 5 Hz, 10 Hz, 25 Hz) of the dynamic modulus of two typical asphalt (see Table 2 and Table 3), fitting the dynamic modulus master curve of asphalt mixture at an analysis temperature of 20°C (see Fig 2 and Fig 3), in order to describe the viscoelastic mechanical properties of asphalt surface layer. The

**Table 2. Asphalt mixture's Dynamic Modulus in the upper layer.**

| Temp(°C) | Dynamic Modulus E*(kPa) | | | | | |
|---|---|---|---|---|---|---|
| | 0.1 Hz | 0.5 Hz | 1 Hz | 5 Hz | 10 Hz | 25 Hz |
| 5 | 8927000 | 11171000 | 12109000 | 14139000 | 14933000 | 15897000 |
| 20 | 3519000 | 4727000 | 5525000 | 7595000 | 8549000 | 9830000 |
| 35 | 886000 | 1437000 | 1768000 | 2803000 | 3376000 | 4425000 |
| 50 | 302000 | 456000 | 553000 | 886000 | 1091000 | 1438000 |

**Table 3. Asphalt mixture's Dynamic Modulus in the lower layer.**

| Temp(°C) | Dynamic Modulus E*(kPa) | | | | | |
|---|---|---|---|---|---|---|
| | 0.1 Hz | 0.5 Hz | 1 Hz | 5 Hz | 10 Hz | 25 Hz |
| 5 | 7031000 | 9496000 | 10571000 | 12960000 | 13909000 | 15067000 |
| 20 | 2120000 | 3505000 | 4264000 | 6364000 | 7380000 | 8782000 |
| 35 | 551000 | 963000 | 1228000 | 2127000 | 2660000 | 3515000 |
| 50 | 192000 | 300000 | 372000 | 633000 | 806000 | 1111000 |

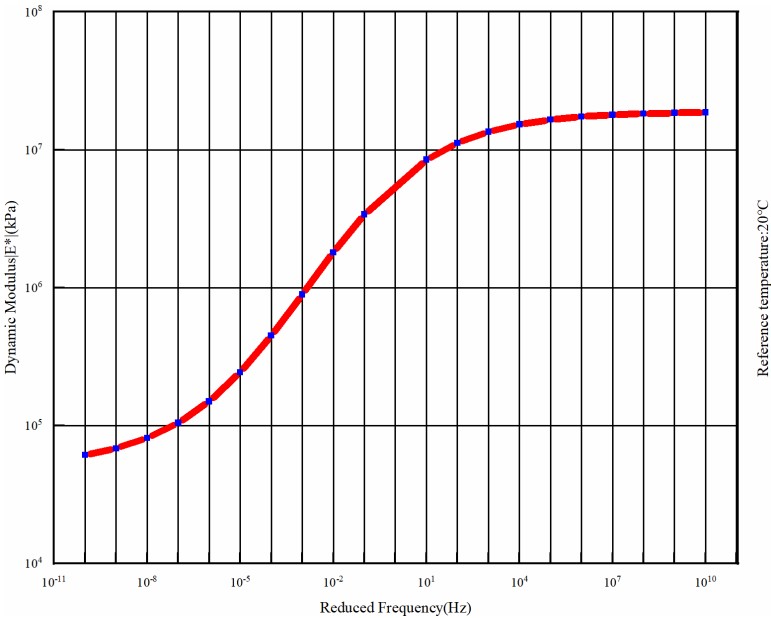

**Fig 2. Master curve of dynamic modulus of the upper layer.**

combination of the dynamic modulus master curve and the time-temperature equivalence factor can reflect the mechanical properties of the viscoelasticity of asphalt mixtures, and can predict and analyze the mechanical properties of the material under extreme conditions such as low temperatures and high frequencies, which are difficult to obtain through tests [29].

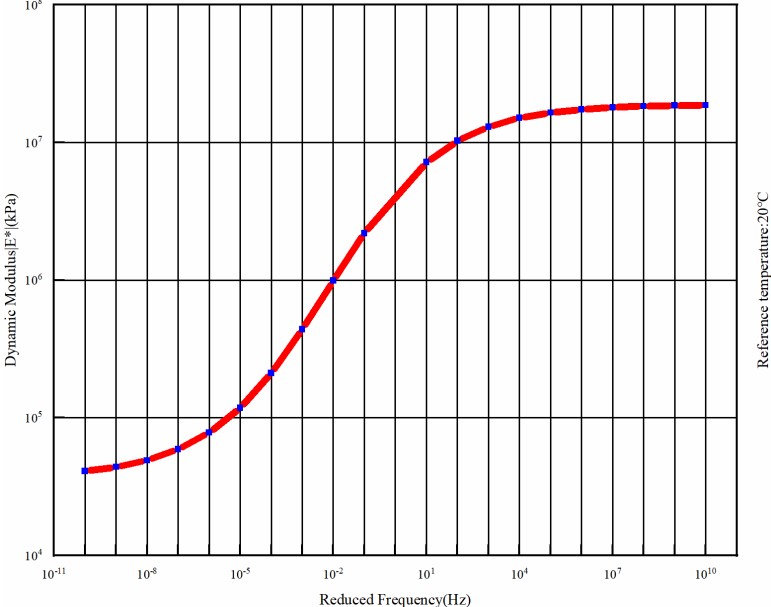

**Fig 3. Master curve of dynamic modulus of the lower layer.**

The Poisson's ratio of the upper and lower layers was taken as 0.25, the density of the upper layer was 24 kN/m³, the density of the lower layer was 23.5 kN/m³, and the damping ratio was taken as 10%.

**3.4.2 Parameters of recycled base.** Typically, a subgrade with cement stabilized aggregates is called a semi-rigid subgrade, but in reclaimed pavements due to in-situ crushing of materials in addition to reclaimed aggregate or reclaimed inorganic binder stabilized aggregate (RAI), there may be materials from reclaimed asphalt pavement (RAP). The mixing of asphalt materials also results in a smaller modulus of the reclaimed subgrade than a normal semi-rigid subgrade when cement is used as the stabilizer recycled material.

However, due to the existence of reclaimed aggregate or reclaimed inorganic binder stabilized aggregate (RAI), its modulus is larger than that of the flexible subgrade stabilized with asphalt. Therefore, in this paper, the modulus of cement stabilized recycled subgrade is located between the flexible subgrade and semi-rigid subgrade, and its size depends on the content of RAP in recycled material, but RAP/RAI should be less than 40%.

Therefore, in this paper, the modulus of recycled base layer is taken as 8000 MPa, Poisson's ratio is taken as 0.25, the thickness of recycled base layer is taken in the range of 24 cm, and the density is taken as 22.5 kN/m³.

**3.4.3 Parameters of old road residual layer.** The old road residual layer is the pavement structure remaining after breaking up some or all of the old road subgrade, which includes the roadbed portion of the original pavement and which serves as the roadbed in the regenerated pavement structure. Its thickness may be regarded as an infinite semi-space structure. The residual layer modulus of the old road ranges from 150 MPa. poisson's ratio is taken as 0.4. Combined with the above expressions the structural parameters of the pavement in this paper are summarized in Table 4.

## 3.5 Response point setting

For the single-axle double-wheel set load, the double-wheel load calculation points and vertical locations are shown in Fig 4. In calculating the structural response of the pavement, the maximum mechanical response at four locations, A, B, C and D, is generally selected as a parameter [30], where point D is the mid-point location of points B and C. In the 3D

**Table 4. Parameters related to pavement structure.**

| Layers | Materials | Modulus (MPa) | Poisson's ratio | Damping ratio | density (kN/m³) |
|---|---|---|---|---|---|
| Asphalt | AC-13 | – | 0.25 | 10% | 24 |
| Asphalt | AC-20 | – | 0.25 | 10% | 23.5 |
| Base | RAI、RAP | 8000 | 0.25 | 0 | 22.5 |
| Subgrade | old road material | 150 | 0.4 | 0 | 19 |

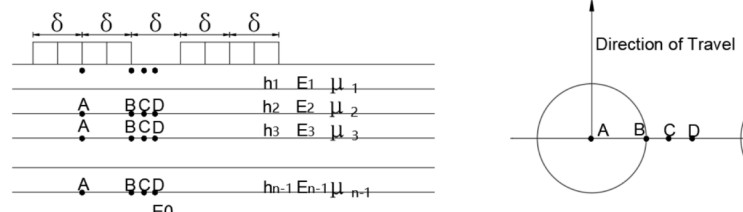

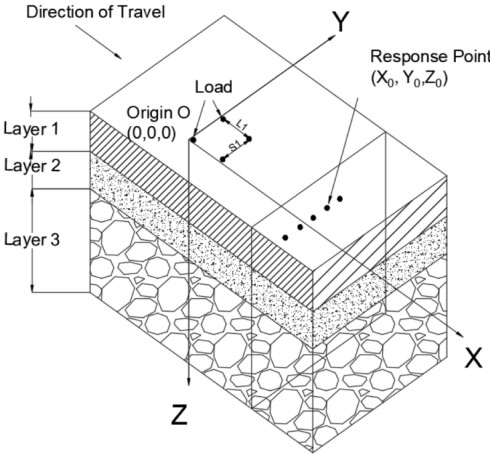

**Fig 4. Two-wheel load response point location and vertical position.**

model, the X-axis represents the direction of travel, the Z-axis represents the direction along the depth, and the Y-axis is perpendicular to the X and Z axes.

## 4 Result and discussion

### 4.1 Temporal analysis

**4.1.1 Temporal change of vertical displacement at the top of asphalt surface layer.** From the Fig 5, it can be seen that the vertical displacement changes of the four points ABCD have the same trend, so only the time-range changes of point A are given here graphically.

From Fig 5(a), the overall trend of the time-range change of vertical displacement at point A is increasing from 0 and then decreasing to 0, with the peak value appearing at about 0.18s. This phenomenon can be attributed to the dynamic interaction between the wheel load and the pavement structure. When the wheel first approaches point A, the pavement

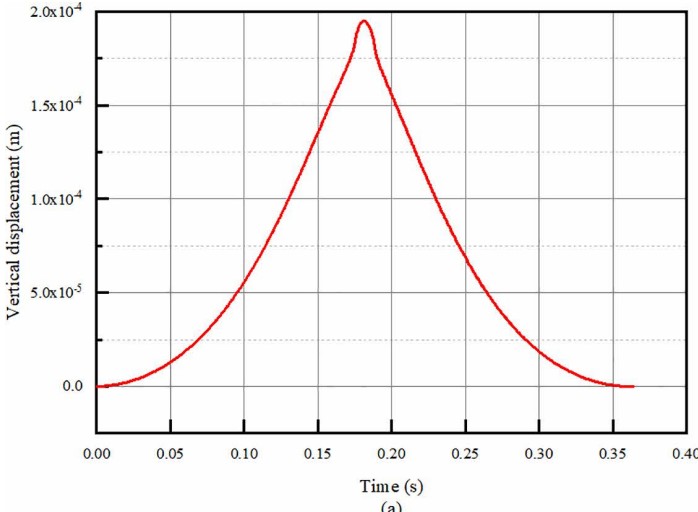
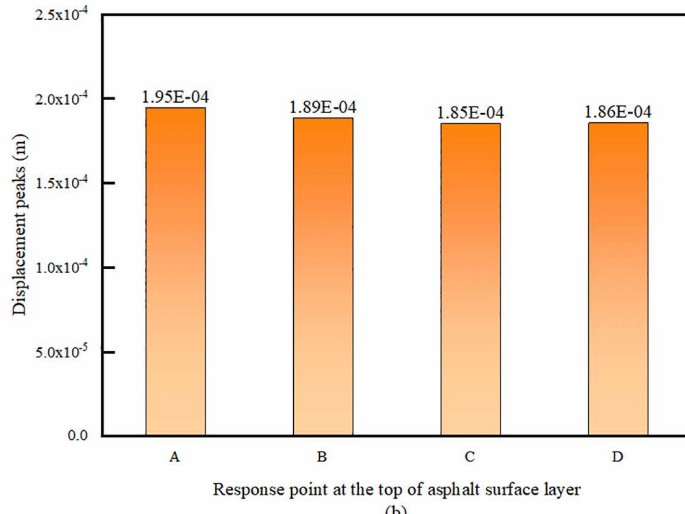

**Fig 5. Simulation results of vertical displacement at the top of asphalt layer.** (a) Vertical displacement temporal change of point A, (b) Comparison of vertical displacement peaks at each response point.

begins to deform under the load, causing the vertical displacement to increase. As the wheel passes, the elastic - like properties of the pavement materials allow the structure to gradually recover, leading to the decrease in vertical displacement back to 0. From Fig 5(b), the peak value of the vertical displacement of the road surface becomes smaller as the distance from each response point to the center of the wheel track increases. The underlying mechanism is related to the stress distribution law. The wheel load exerts the greatest influence on the area directly beneath it. As the distance from the wheel track center increases, the stress is distributed over a larger area of the pavement structure, resulting in a smaller vertical displacement at these points. And the difference of the peak vertical displacement of the four points is not big, the peak difference between point A and points B and D is 3.3% and 5.0%, and the peak displacement of point C, which is the smallest one, is only 5.2%.

This indicates that, in terms of vertical displacement characteristics, these points are relatively similar in response to the wheel load. Therefore, it is sufficient to refer to point A when studying the vertical displacement hereafter, and the three points BCD can be left out of the reference.

**4.1.2 Temporal change of stress and strain in X and Y directions of asphalt overlay layer bottom.** The four points A, B, C, and D in Fig 6 show roughly the same trend of X-directional stresses, all of which show positive and negative bimodal peaks. The corresponding peaks of compressive stresses all appeared near 0.18s, while the tensile stresses were very small, so they were neglected.

For the X-direction stress, when the moving load is driven at a uniform speed of 60 km/h, a transient smaller tensile stress will appear in the overlay layer bottom. The front part of the wheel may cause a slight stretching effect on the pavement material in the X-direction in the overlay layer bottom. However, the overall compressive stress is dominant because the main force exerted by the wheel on the pavement is in the vertical direction, which is then transferred and redistributed in the X-direction, leading to a much larger compressive stress. The relatively small value of the tensile stress compared to the compressive stress makes its impact on the pavement's mechanical response negligible.

From the temporal change graph, it can be seen that the tensile stress in the X direction is very small, and the effect on the pavement is negligible, so there is no statistical peak tensile stress.

From the peak stress of the overlay layer bottom, the X-directional stress of the overlay layer bottom decreases with the increase of the distance from the response point to the center of the wheel track, i.e., the X-directional stress at point

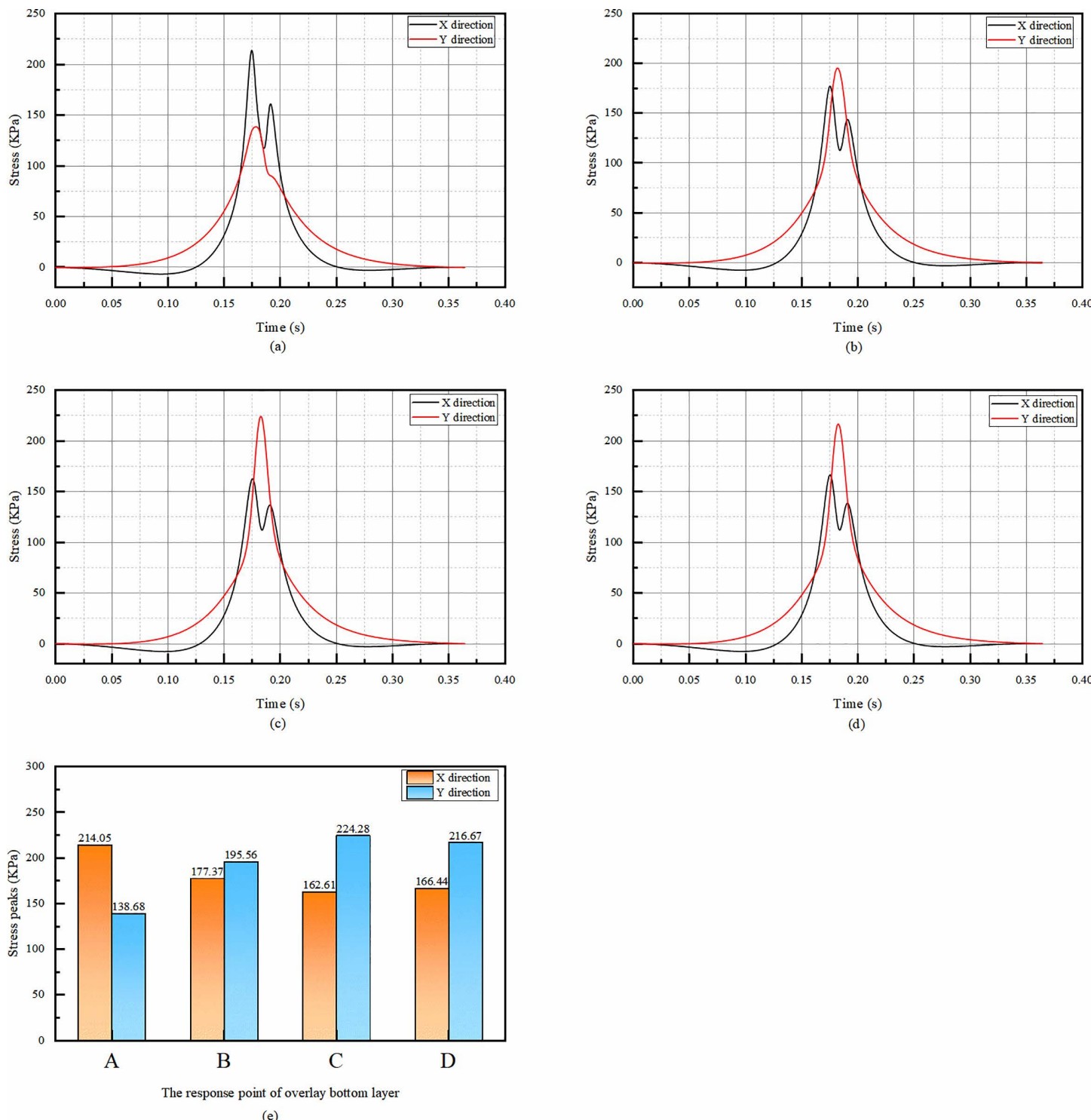

**Fig 6. Simulation results of bottom stress of asphalt overlay layer.** (a)The temporal change of X and Y direction stress at point A, (b)The temporal change of X and Y direction stress at point B, (c)The temporal change of X and Y direction stress at point C, (d)The temporal change of X and Y direction stress at point D, (e)Comparison of stress peaks in X and Y directions at each response point.

A is the largest among the four calculation points; while the peak Y-directional stress increases with the increase of the distance from the calculation point to the center of the wheel track, i.e., the Y-directional stress at point C is the largest among the four response points.

As can be seen from the Fig 7, from the viewpoint of temporal change, the X-direction strain of the overlay layer bottom has the same trend, both of which are slowly appearing tensile strain at first, but the tensile strain is very small and negligible, and then the peak of compressive strain occurs rapidly. The rapid occurrence of the compressive strain peak is because the vertical load from the wheel is transferred and redistributed in the X-direction, causing significant compression of the pavement material. After reaching the peak, the strain decreases but does not change from compressive strain to tensile strain. Subsequently, the compressive strain increases again and reaches the second positive peak, but the second positive peak is significantly smaller than the first peak. The subsequent increase in compressive strain to form a second, smaller positive peak is likely due to the continuous movement of the load and the visco-elastic properties of the pavement materials. The visco-elastic nature allows the material to respond differently as the load passes, resulting in this secondary compression effect. Since the first positive peak represents the maximum deformation under the load's main action. Therefore, only the first positive peak needs to be considered subsequently to see if it meets the requirements.

The fact that the Y-direction strain is compressive throughout the temporal change indicates that the pavement structure has a higher resistance to deformation in this direction under the applied load. The significant difference in the temporal change of Y-direction strain at point A, with a double - peak pattern similar to the X-direction, may be related to the specific location of point A relative to the wheel track and the overall pavement structure.

From the Fig 7(e), the X-direction strain peak at point A is the largest among the four points, and only the strain at point A needs to be taken into account in the subsequent consideration of the design.

**4.1.3 The X and Y direction stress and strain temporal changes of the bottom of the recycled base layer.** As can be seen from the Fig 8, the temporal changes of X- and Y-directional stresses in the bottom of the recycled base layer are in the shape of "basin".

The peaks of X-direction stress of recycled base layer are positive and negative double peaks, in the whole range of time, there is a tensile and compressive transformation, graphically the tensile stress is much larger than the compressive stress, and the peaks of Y-direction stress are single peaks, all of them are negative peaks, and there is only the compressive stress in the whole range of time. the negative peaks of X-direction stress are larger than the negative peaks of Y-direction stress, so for the comparison of the peaks, the peaks of the X-direction stress are only compared with the peaks of the response points of the X-direction stress.

As seen from Fig 8(e), the peak of X-directional tensile stress at point A is the smallest, and the peaks of stresses at points B, C, and D are closer to each other, with a difference of 3.3% between the maximum compressive stress and the minimum compressive stress, and a difference of 5.5% between the maximum tensile stress and the minimum tensile stress, which are all within the error range. This may be because the load dispersion mechanism of the recycled base layer effectively and uniformly distributes stress in the area. It can be considered that the response point position has little influence on the stress peak value can be ignored, and the follow-up can only study point A.

From the Fig 9, the temporal changes of X- and Y-direction strains of recycled base layer at the four calculation points are in the shape of "basin" as the temporal changes of stress. The X-direction strain has double peaks, positive and negative, in the whole range of time course, but the tensile strain is much larger than the compressive strain. The relatively large tensile strain is due to the dynamic behavior of the regenerated substrate under wheel load. When the load approaches, the material in the recycled base layer begins to deform. The lateral expansion of the load leads to initial tension in the X direction, resulting in a peak tensile strain. And the Y-direction strain has only a single peak, negative, in the whole range, only tensile strain, and the peak value of Y-direction tensile strain at the four points is smaller than that of X-direction tensile strain. So the comparison of the peaks is only comparing the peaks of each response point of X-direction strain.

 

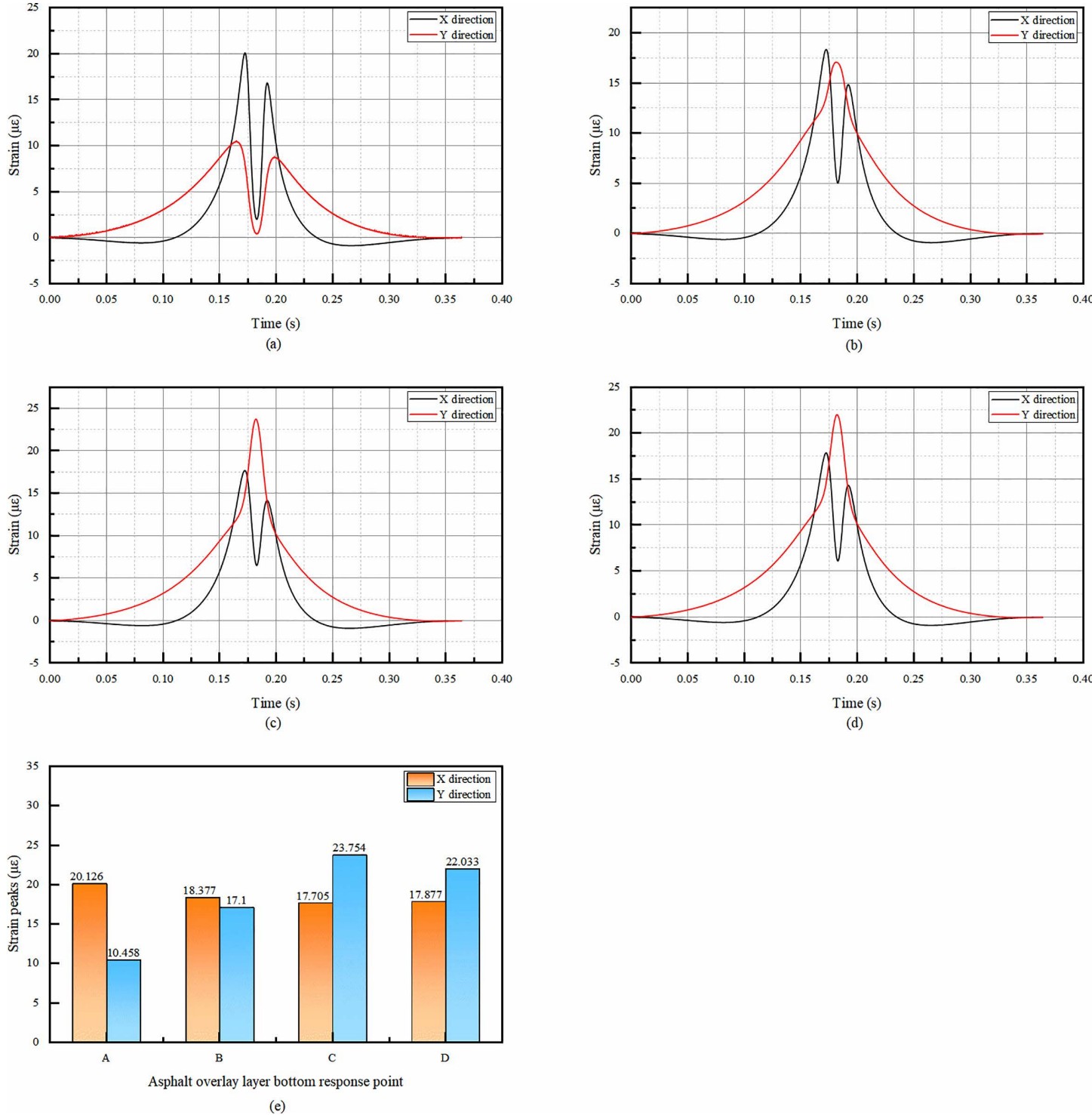

**Fig 7. Simulation results of bottom strain of asphalt overlay layer.** (a)The temporal change of strain in X and Y directions at point A, (b)The temporal change of strain in X and Y directions at point B, (c)The temporal change of strain in X and Y directions at point C, (d)The temporal change of strain in X and Y directions at point D, (e)Comparison of compressive strain peaks in X and Y directions at each response point.

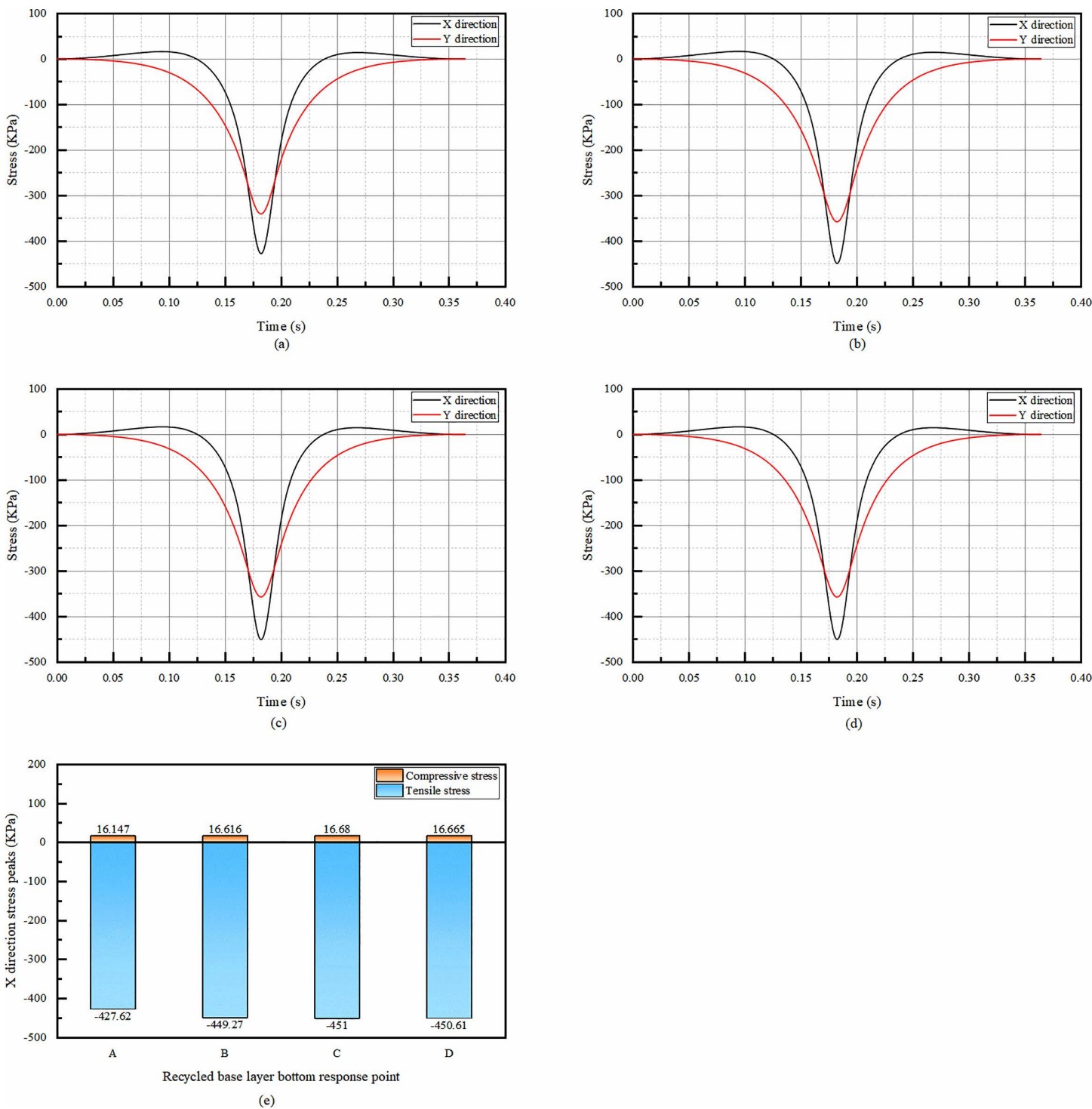

**Fig 8. Simulation results of X-direction stress at the bottom of recycled base layer.** (a) The temporal change of X and Y direction stress at point A, (b) The temporal change of X and Y direction stress at point B, (c) The temporal change of X and Y direction stress at point C, (d) The temporal change of X and Y direction stress at point D, (e) Comparison of X-direction stress peaks at the bottom of recycled base layer.

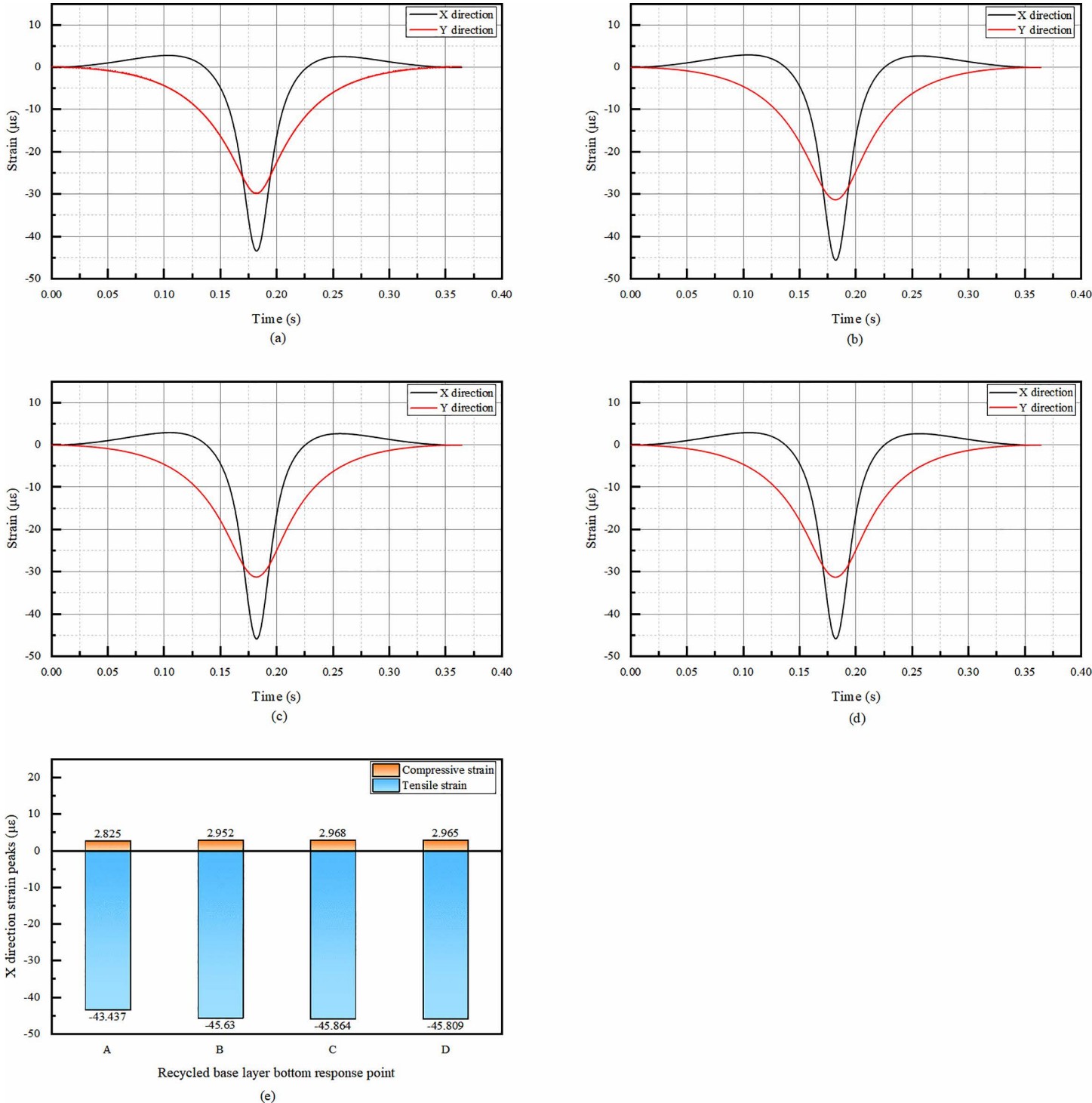

**Fig 9. The simulation results of the bottom strain of the recycled base layer.** (a) The temporal change of X and Y direction strain at point A, (b) The temporal change of X and Y direction strain at point B, (c) The temporal change of X and Y direction strain at point C, (d) The temporal change of X and Y direction strain at point D, (e) Comparison of X-direction strain peaks at the bottom of recycled base layer.

From the comparison of the peak X-direction strain in Fig 9(e), the peak strain of each response point is not much different. The minimum difference in peak strain between the four response points indicates that the recycled base layer exhibits relatively uniform mechanical behavior in the study area. The difference between the maximum compressive strain and the minimum compressive strain is only 5.0%, while the difference between the maximum tensile strain and the minimum tensile strain is only 5.6%, which are within the error range. It can be considered that the effect of response point position on the strain peak is not significant enough to be ignored, and only point A can be studied subsequently.

**4.1.4 Vertical strain at the top of the subgrade.** From the temporal change Fig 10(a), it can be seen that the whole process is compressive strain did not appear tensile strain. The fact that the entire time-process shows only compressive strain in the top of the subgrade is mainly due to the load-transfer mechanism from the upper layers. When a wheel load acts on the pavement surface, the load is gradually transmitted downwards through the overlay layers and the recycled base layer to the subgrade. The subgrade, as the bottom-most structural layer, mainly bears the vertical component of the load, which results in compression. And the subsequent study of the effect of strain on the top of the subgrade, can only consider the compressive strain.

It is easy to see from the Fig 10 comparing the strain peaks that the vertical strain at the top of the subgrade increases with the increase of the distance from the response point location to the center of the wheel track location, but the difference in the strain magnitude of these four points is not large. The difference between point A, which has the smallest vertical strain peak, and point C, which has the largest, is only 4.7%. It can be assumed that the magnitude of Vertical strain at the top of the subgrade is not very much related to the point location. Subsequently, only point A can be studied.

## 4.2 Spatial analysis

Static analysis often focuses only on the mechanical response information of the wheel load to the moment directly above the calculation point, in order to quantify the mechanical response of each point in the depth range of the pavement, and more comprehensively reflect the force changes of each point of the pavement in the entire wheel load time course, by extracting the eigenvalues of the time course change curve, the complex mechanical response curve is transformed into a specific numerical value, which is easy to analyse and compare. Taking Fig 11 as an example, for the case of tensile

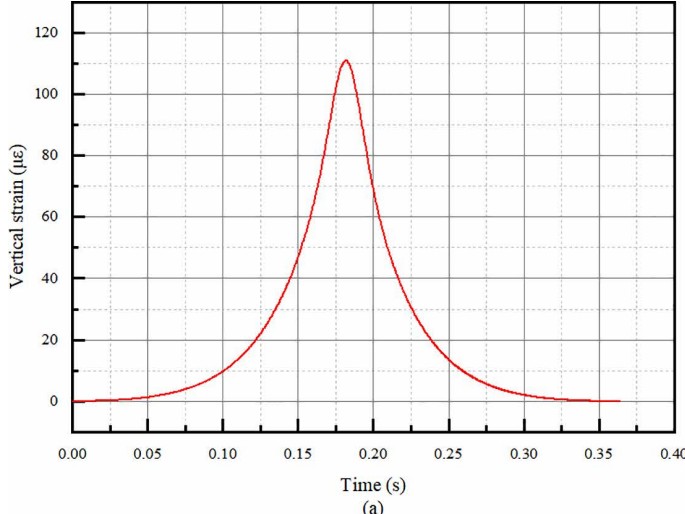
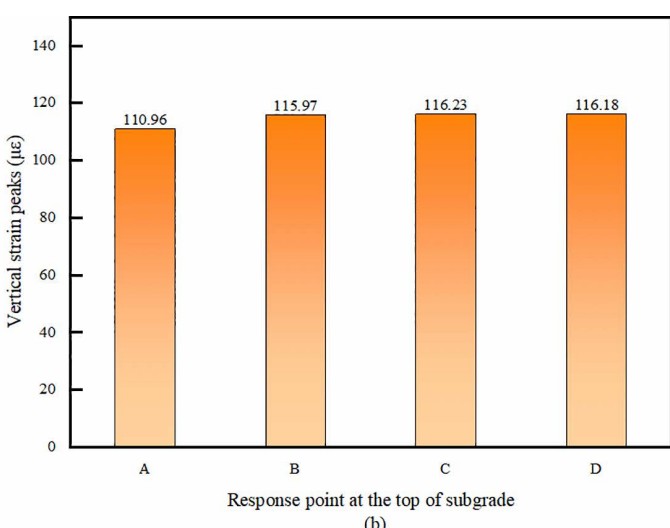

**Fig 10. Vertical strain simulation results of subgrade top.** (a) Vertical strain temporal change of point A, (b) Comparison of vertical strain peaks at each response point.

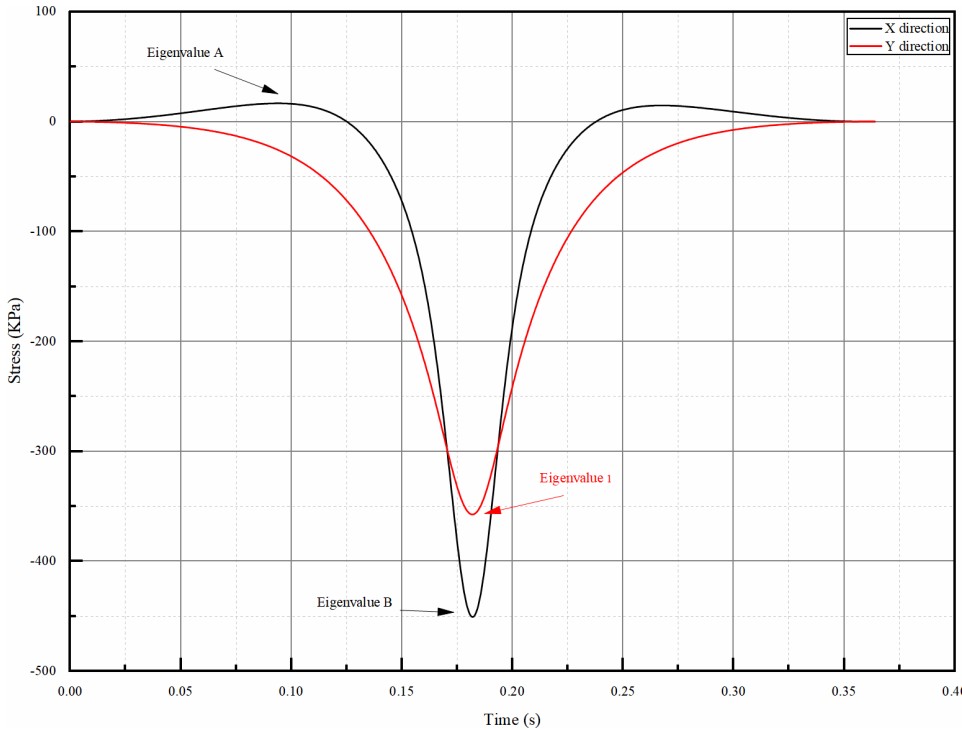

**Fig 11. Example of time curve data extraction graph.**

stress, the eigenvalue 1 can clarify the maximum degree of tensile at the point under wheel load, this is because tensile failure is one of the common failure modes in pavement structure design. By determining the eigenvalue of the maximum tensile degree, the ultimate tensile capacity of the material under wheel load can be evaluated; for the case of both compressive and tensile time course, the eigenvalue A and eigenvalue B indicate the peak value of positive and negative areas respectively. They are chosen as eigenvalues because in the actual force application process on the road surface, the material not only experiences tensile force but also compressive force, and the magnitude and variation of both forces have a significant impact on the fatigue life and overall performance of the road surface. These two eigenvalues can clearly reflect the extreme conditions of the material under different stress states, which is helpful for comprehensively evaluating the performance of the road surface under complex force conditions. Calculation points were selected every 2 cm from the road surface to the top of the residual layer of the old road, and eigenvalues were extracted from the mechanical responses (including vertical displacement, vertical stress, vertical strain, horizontal stress, and horizontal strain) at each point, and plotted the change of each mechanical response with the depth, in order to analyse further the spatial distribution of the mechanical responses.

**4.2.1 Vertical stress distribution along the depth direction.** From the Fig 12, it can be seen that the vertical tensile stress from the top of the asphalt surface layer to the top of the subgrade is almost zero except that it is higher at the top of the asphalt surface layer, and the vertical compressive stress has always existed and its overall value decreases with the increase of the depth, and when it reaches the top of the subgrade it is almost close to zero. The maximum compressive stress occurs at the top position of the asphalt surface layer. Therefore, we do not need to consider the effect of tensile stress, even the top of the asphalt surface layer of tensile stress is very small, can not be considered.

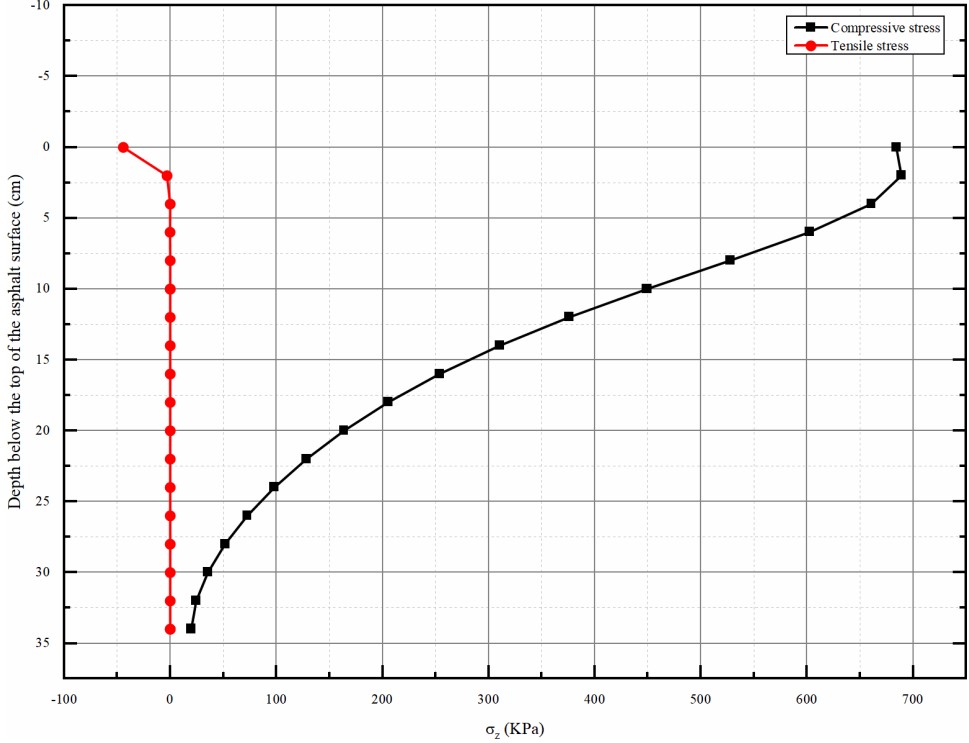

**Fig 12. Vertical stress σz peak depth distribution map.**

**4.2.2 Vertical strain distribution along the depth direction.** From the Fig 13, it can be seen that the vertical tensile stress from the top of the asphalt surface layer to the top of the subgrade is almost zero except that it is higher at the top of the asphalt surface layer, and the vertical compressive stress has always existed and its overall value decreases with the increase of the depth, and when it reaches the top of the subgrade it is almost close to zero. The maximum compressive stress occurs at the top position of the asphalt surface layer. Therefore, we do not need to consider the effect of tensile stress, even the top of the asphalt surface layer of tensile stress is very small, can not be considered.

**4.2.3 Vertical displacement distribution along the depth direction.** As can be seen from the Fig 14, the peak vertical displacement gradually decreases with the increase of depth, and there is an obvious decreasing trend. And the reduction speed is faster before 10 cm, and the reduction trend of the peak vertical displacement slows down when entering the range of subgrade after 10 cm. In the figure, a small mutation of the peak displacement occurs at the bottom of the subgrade and the top of the subgrade.

**4.3 Speed analysis**

**4.3.1 Temporal change of vertical displacement at the top of asphalt pavement with different speeds.** As can be seen from Fig 15(a), the trend of the temporal change of the vertical displacement at point A under different velocities is basically the same, only there are differences in the peak size and time.

From Fig 15(b) it can be seen that the faster the speed, the smaller the peak vertical displacement at point A. This may be related to the time of the load, the faster the speed through the point A the shorter the time, the displacement has not yet occurred completely the load effect has ended.

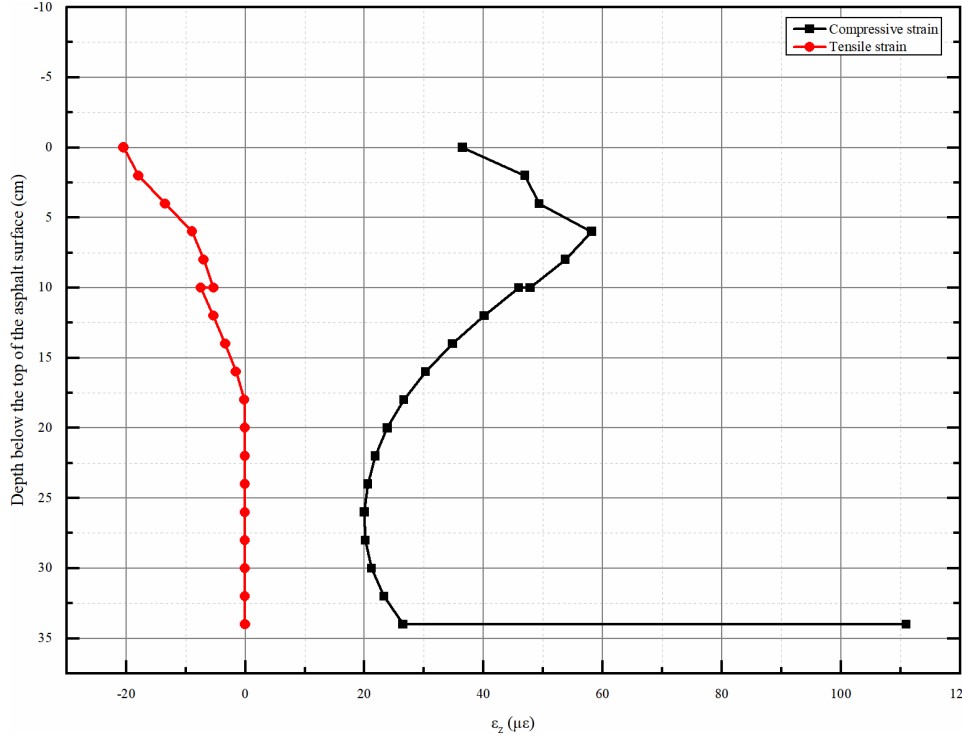

**Fig 13. Vertical strain εz peak depth distribution map.**

However, the difference of vertical displacement of the top of asphalt surface layer under each speed is very small, the vertical displacement under 20km/h and 80km/h only differ by 4.4%. The maximum vertical displacement of the asphalt surface layer under dynamic is 11.3% different from that under static. Therefore, it can be concluded that the degree and time of deformation suffered by the top of the asphalt surface layer decreases as the speed increases.

**4.3.2 Temporal change of X-direction strain in asphalt overlay layer bottom at different speeds.** From the temporal change plots of X-direction strain in the overlay layer bottom of asphalt, it can be seen that the strain peaks are positive and negative bimodal peaks for all velocities. Combined with Fig 16(b), it can be seen that the peaks of both bimodal peaks tend to decrease as the speed increases.

From the temporal change, it is seen that the tensile strain appears first in the whole process, and then the compressive strain rises sharply and reaches the peak value in a very short period of time. After reaching the peak value, the compressive strain decreases sharply but does not disappear completely, and then rises again to reach the second peak value, but the second peak value is obviously smaller than the first peak value. Therefore, it is sufficient to consider only the first peak in the design.

After reaching the second peak, the compressive strain decreases sharply and becomes tensile strain, and then the trend slows down and decreases to 0. From the comparison of the peak values in Fig 16(b), the peak values of compressive and tensile strains decrease with the increase of speed.

The X-directional tensile strain of the asphalt overlay layer bottom under static condition is 0.08109 με, which is a big difference compared with the tensile strain at each speed.

**4.3.3 Temporal change of X-direction stress at the bottom of recycled base layer with different speeds.** The temporal change of X-direction stress of recycled base layer is in the shape of "basin", and the trend of temporal change

**Fig 14. Vertical displacement Sz peak depth distribution map.**

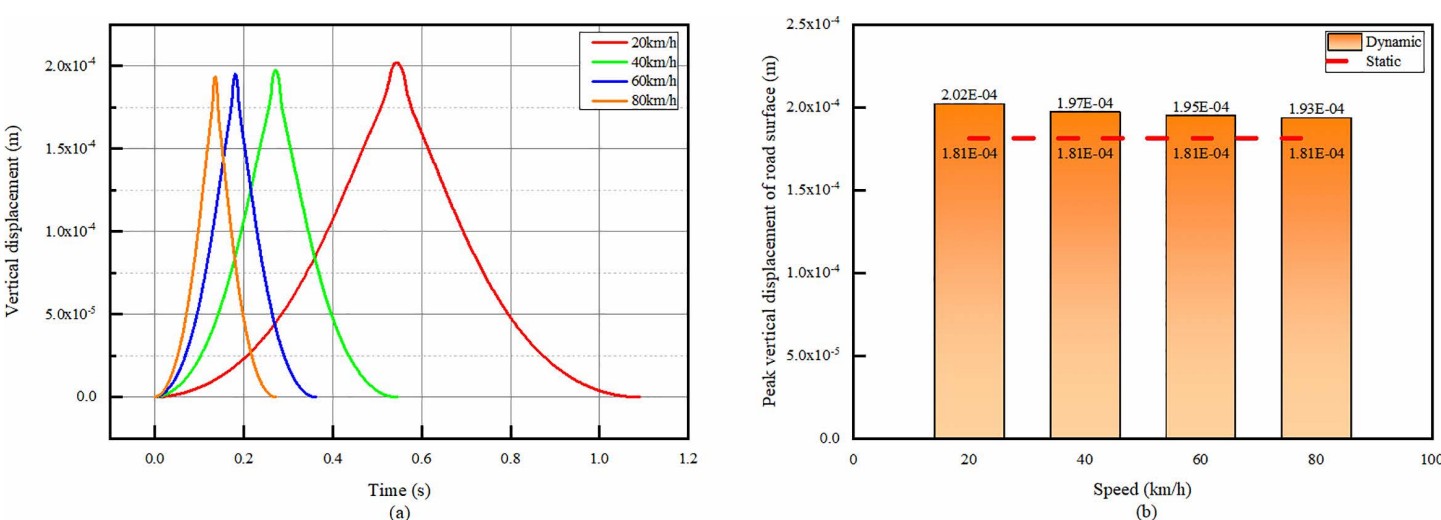

**Fig 15. Vertical displacement at the top of asphalt layer.** (a)The temporal change of vertical displacement, (b)Peak vertical displacement comparison.

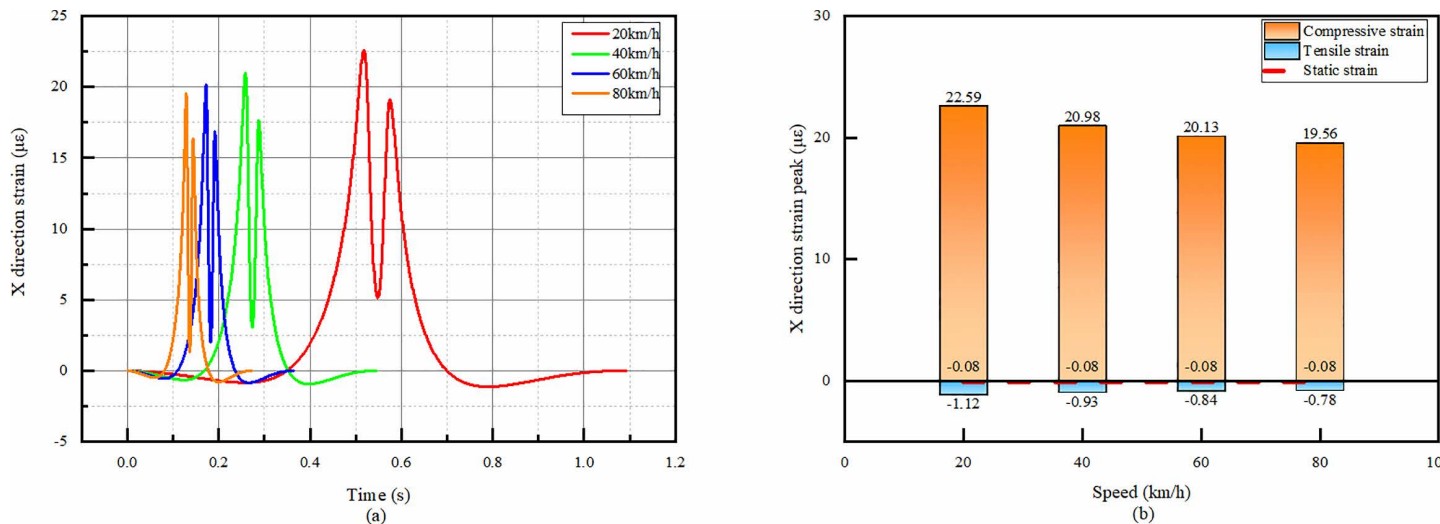

**Fig 16. X-direction strain simulation results of asphalt overlay layer bottom.** (a)The temporal change of X-direction strain, (b)Peak strain comparison.

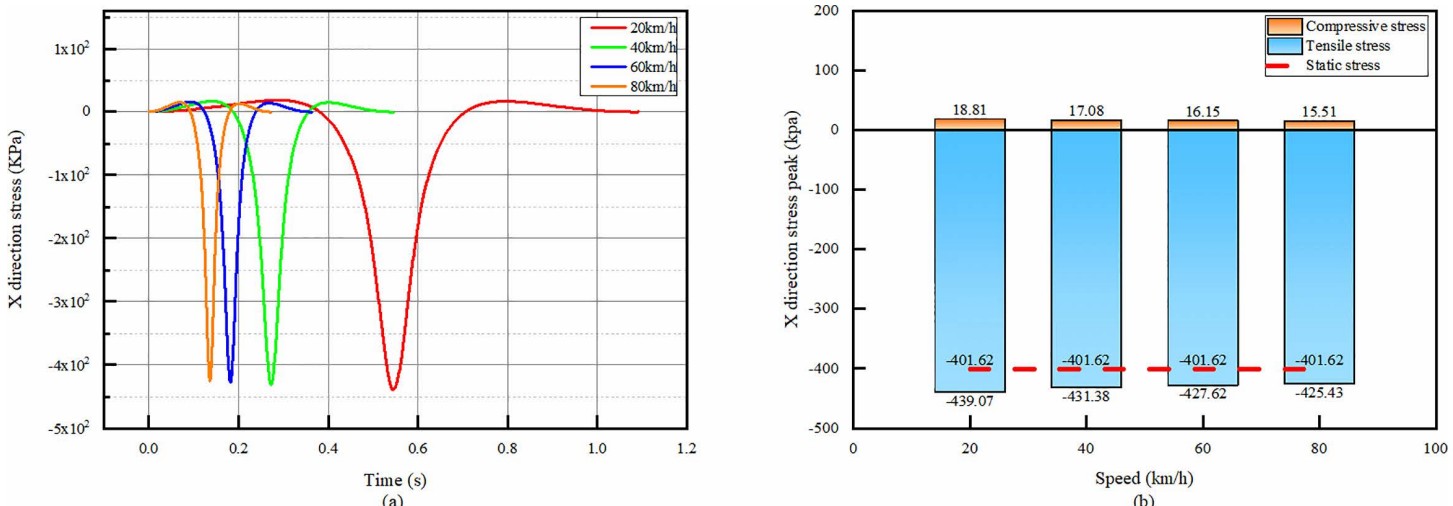

**Fig 17. Simulation results of X-direction stress at the bottom of recycled base layer.** (a)The temporal change of X-direction stress, (b)Peak stress comparison.

of stress is basically the same in four speeds, and there are positive and negative double peaks, i.e., compressive and tensile stresses appeared in the whole temporal change. From the comparison of stress peaks, with the increase of speed, the X-direction compressive stress and tensile stress at the bottom of the recycled base layer show a decreasing trend. However, the compressive stress is very small can be disregarded, and the difference between the maximum and minimum values of tensile stress is only 3.2%. And when comparing with the stress under static, the tensile stress under static is obviously smaller than dynamic. The difference between the tensile stress under static and the maximum tensile stress under dynamic is 9.3%.

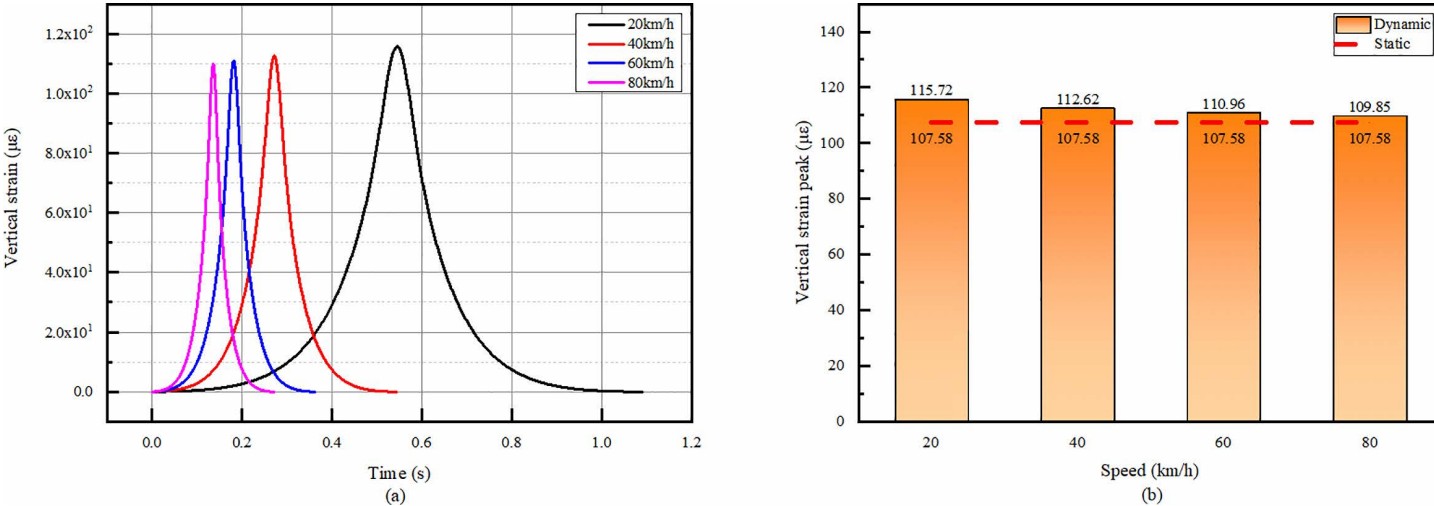

**Fig 18. Vertical strain calculation results of old road residual layer.** (a)The temporal change of vertical strain, (b)Peak strain comparison.

**4.3.4 Vertical strain at the top of the residual layer of the old road at different speeds.** As shown in the Fig 18, the trend of the vertical strain time course of the residual layer of the old road under different velocities is the same, and there is only compressive strain but no tensile strain in the whole, and the strain peak is reached in the middle part of the time course. The peak strain gradually decreases with the increase of speed, and the maximum strain is 5.3% larger than the minimum strain.

## 4.4 Temperature analysis

**4.4.1 Temporal change in vertical displacement at the top of asphalt surface at various temperatures.** From the temporal changes in vertical displacement at the top of the asphalt surface at each temperature, it can be seen that the

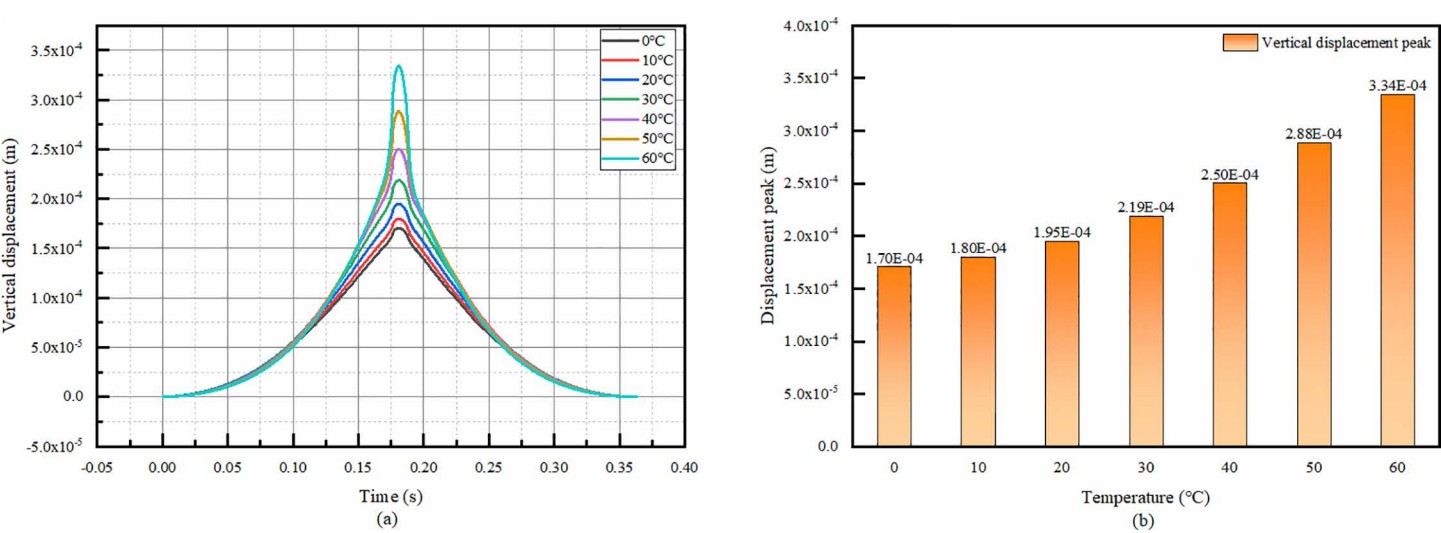

**Fig 19. Vertical displacement at the top of asphalt layer.** (a)The temporal change of vertical displacement, (b)Peak vertical displacement comparison.

loading times are all the same since the velocities are the same. All of them reach the peak value around 0.18s. That is, the deformation is maximum at this time. From the comparison of the peak vertical displacement, it can be seen that the vertical displacement of the top of the asphalt surface gradually becomes larger as the temperature increases. That is, the higher the temperature, the greater the deformation of the top of the asphalt surface. the vertical displacement of 0°C and 60°C difference of 96%, which shows that the temperature on the top of the asphalt surface of the vertical displacement has a great influence.

### 4.4.2 Temporal change of X-direction strain in asphalt overlay layer bottom at various temperatures.

The temporal changes show that there are both positive and negative bimodal peaks in the X-direction strain of the asphalt overlay layer bottom, although the shapes of the images change significantly at different temperatures. In particular, the positive peak of compressive strain is reached at 0°C and 10°C, after which the strain decreases rapidly and converts to tensile strain and reaches the minimum value. At other temperatures, although the strain tends to decrease after the positive peak, it does not change into tensile strain, and the peak of tensile strain occurs in the second half of the image.

It can be seen that the temperature has a great influence on the X-direction strain of the overlay layer bottom. From the Fig 20, with the increase of temperature, the compressive strain is gradually larger, and the tensile strain decreases first and then increases.

### 4.4.3 Temporal change of X-direction stress at the bottom of recycled base layer at various temperatures.

From the shape of the temporal change of the X-direction stress at the bottom of the recycled base layer, the shapes are all similar, and there are obvious positive and negative bimodal peaks, which means that compressive and tensile stresses occur throughout the process. The peak of tensile stress is significantly greater than compressive stress. In terms of peak comparison, both tensile and compressive stresses increase with increasing temperature, and the minimum value of tensile stress is 47% smaller than the maximum value.

Obviously, the performance of the base layer is greatly affected as the temperature increases, making it more susceptible to damage.

### 4.4.4 Vertical strain at the top of the residual layer of the old road (top of the subgrade) at each temperature.

From the view of temporal change, the trend of vertical strain change at the top of the residual layer at

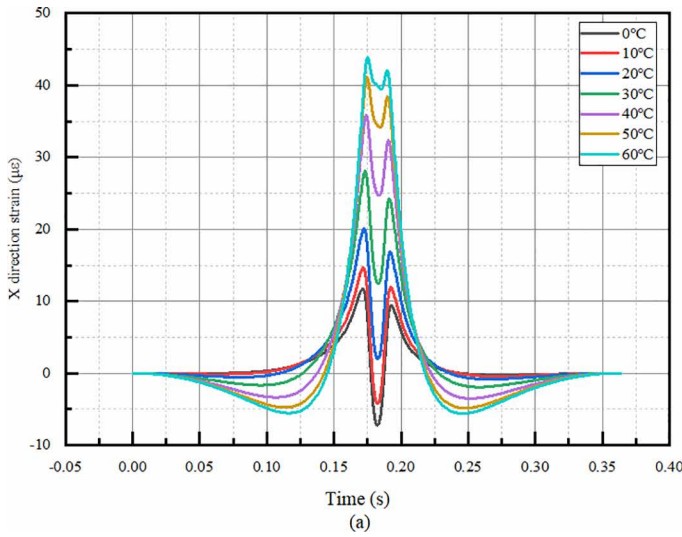
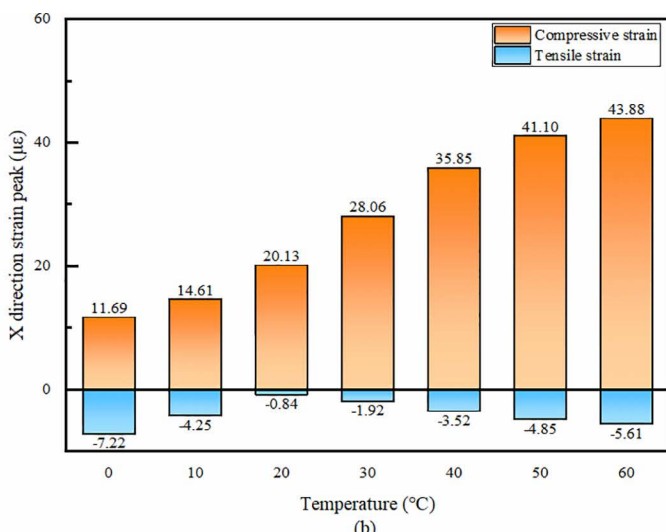

**Fig 20. X-direction strain simulation results of asphalt overlay layer bottom. (a)The X-direction strain temporal change, (b)Peak strain comparison.**

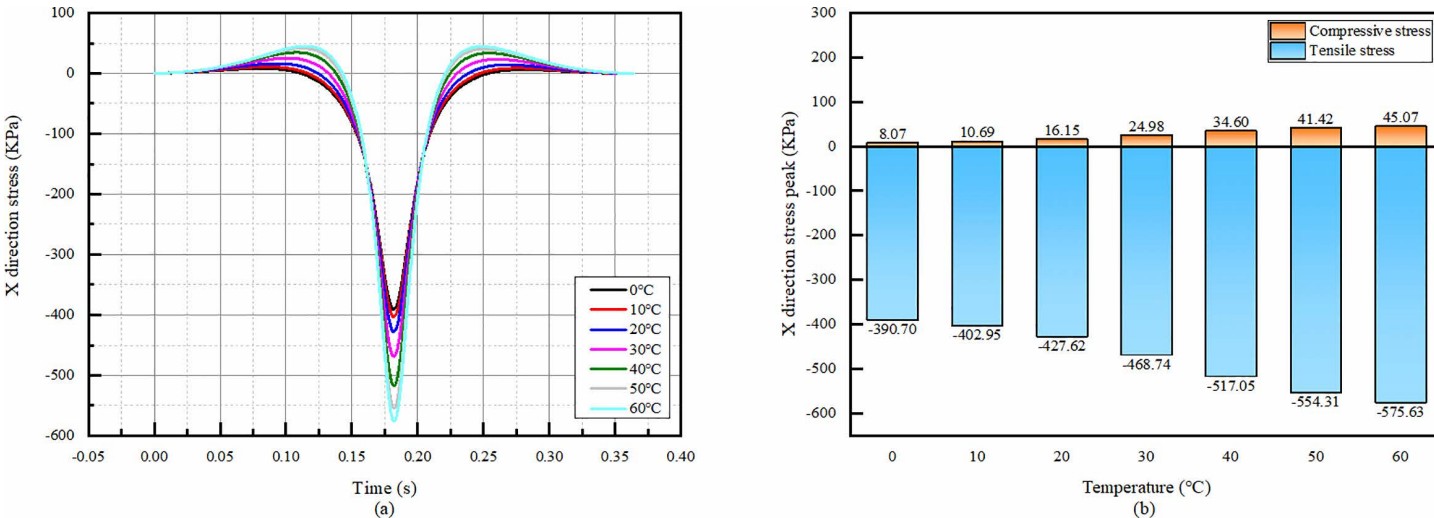

**Fig 21. Simulation results of X-direction stress at the bottom of recycled base layer.** (a) The temporal change of X-direction stress, (b) Peak stress comparison.

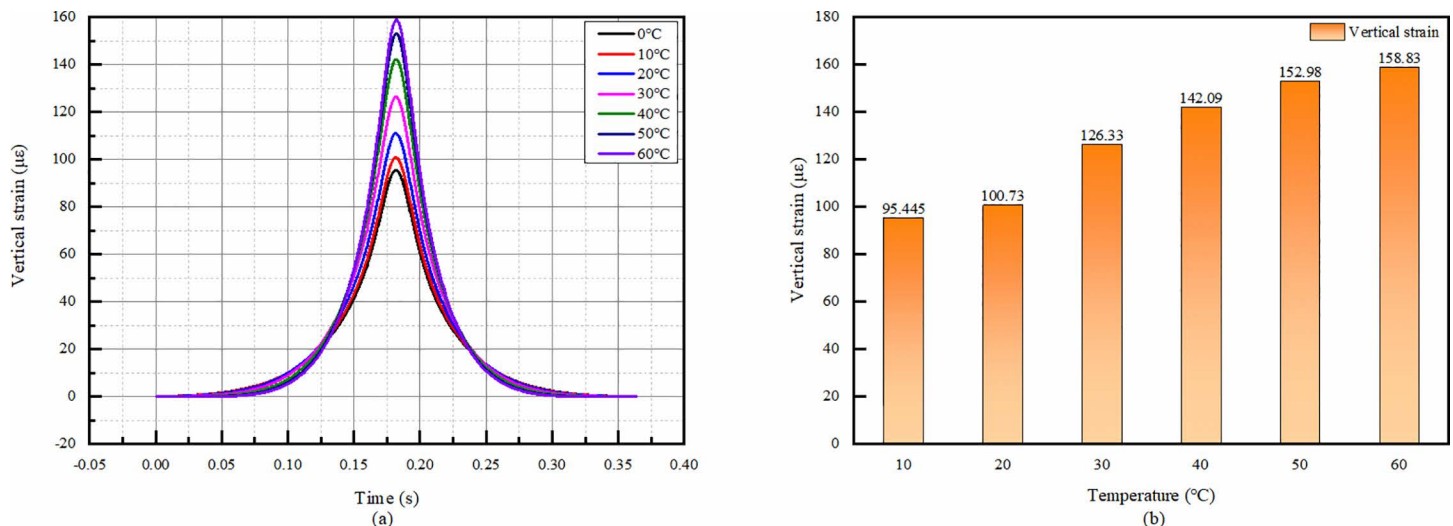

**Fig 22. Vertical strain calculation results of old road residual layer.** (a)The vertical strain temporal change, (b)Peak vertical strain comparison.

different temperatures is the same, and all of them reach the peak value around 0.18s, and there is no negative value in the whole temporal change, which are all compressive strains. Therefore, we do not have to consider the influence factor of tensile strain when carrying out the design.

From the comparison of the peak value, the vertical compressive strain at the top of the residual layer increases gradually with the increase of temperature, and the peak strain at 10°C is 66% smaller than that at 60°C. That is to say, temperature has a great influence on it, and higher temperature will cause larger compressive strain thus leading to the destruction of pavement material.

## 5 Conclusion

In this paper, relying on the typical recycled pavement structure, the structural model of 10(4＋6)-8000(24 cm)-150(thickness of surface layer 10 cm(4 cm＋6 cm),modulus of base layer 8000 MPa, modulus of residual layer 150 MPa) is taken as a representative, and the 3D-Move Analysis is applied to construct the full-depth reclaimed Pavement numerical model for dynamic analysis to explore the mechanical properties of FDR-PC reclaimed pavement by plane, space, speed and temperature, the main conclusions of this study are shown below:

1. From the plane aspect, the mechanical response values of vertical displacement at the top of asphalt surface layer, X-directional strain at the overlay base layer bottom, X-directional tensile stress at the bottom of recycled base layer and vertical strain at the old road residual layer are analyzed by comparing the four computational points ABCD of the structural layer. The difference between these four points is not significant, so only point A can be studied in the subsequent study of these four mechanical responses.

2. From the spatial aspect, the peak vertical tensile stress along the depth direction from the top of the asphalt surface to the top of the residual layer of the old road decreases with the increase of depth. The maximum vertical strain along the depth direction occurs at the top of the old road remnant layer, which can be a design indicator. Vertical displacements along the depth direction as a whole continue to decrease with increasing depth.

3. In terms of speed, within the speed range of 20 km/h to 80 km/h, as speed increases, the displacements, stresses and strains affecting the pavement structure are reduced accordingly, and the time of action is shortened. Therefore, within this speed range, the faster the traveling speed of the pavement vehicle, the smaller the impact.

4. From the aspect of temperature, with the increase of temperature, each mechanical response are different degrees of increase, that is to say, the deformation damage to the pavement will be more serious.

Looking forward to the future, we can further deepen our understanding of recycled pavement from the following research directions. It is a potential research field to explore the influence of the mix proportion and construction technology of different recycled material mixtures on the mechanical properties of pavement. With the increasing emphasis on sustainable infrastructure construction, the role of full depth recycled pavement has attracted more and more attention. By studying this, it will contribute to the wide adoption and better performance of environmentally friendly pavement solutions.

## Supporting information

**S1 Fig. Graphical abstract.**
(TIF)

## Author contributions

**Conceptualization:** Haiwei Zhang, Ning Liu, Bowei Sun.

**Data curation:** Haiwei Zhang, Ning Liu.

**Formal analysis:** Haiwei Zhang, Ning Liu, Qingqing Zhang.

**Funding acquisition:** Haiwei Zhang.

**Investigation:** Haiwei Zhang, Ning Liu, Qingqing Zhang, Jiazhen Liu.

**Methodology:** Haiwei Zhang, Ning Liu, Bowei Sun.

**Project administration:** Haiwei Zhang.

**Resources:** Haiwei Zhang, Ning Liu.

**Software:** Haiwei Zhang, Ning Liu, Qingqing Zhang, Jiazhen Liu.

**Supervision:** Haiwei Zhang.

**Validation:** Haiwei Zhang, Ning Liu, Bowei Sun.

**Visualization:** Haiwei Zhang, Ning Liu.

**Writing – original draft:** Haiwei Zhang, Ning Liu.

**Writing – review & editing:** Haiwei Zhang, Ning Liu, Bowei Sun.

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
