## [Decision Letter · Decision Letter 0]

11 Mar 2025

PONE-D-25-09395Mechanical response characteristics of Full Depth Reclaimed Pavement based on 3D-Move modelPLOS ONE

Dear Dr. Zhang,

Thank you for submitting your manuscript to PLOS ONE. After careful consideration, we feel that it has merit but does not fully meet PLOS ONE’s publication criteria as it currently stands. Therefore, we invite you to submit a revised version of the manuscript that addresses the points raised during the review process.

We look forward to receiving your revised manuscript.

Kind regards,

Jiaolong Ren

Academic Editor

PLOS ONE

3. In the online submission form you indicate that your data is not available for proprietary reasons and have provided a contact point for accessing this data. Please note that your current contact point is a co-author on this manuscript. According to our Data Policy, the contact point must not be an author on the manuscript and must be an institutional contact, ideally not an individual. Please revise your data statement to a non-author institutional point of contact, such as a data access or ethics committee, and send this to us via return email. Please also include contact information for the third party organization, and please include the full citation of where the data can be found.

Reviewers' comments:

Reviewer's Responses to Questions

**Comments to the Author**

1. Is the manuscript technically sound, and do the data support the conclusions?

Reviewer #1: Yes

Reviewer #2: Yes

2. Has the statistical analysis been performed appropriately and rigorously? 

Reviewer #1: Yes

Reviewer #2: Yes

3. Have the authors made all data underlying the findings in their manuscript fully available?

Reviewer #1: Yes

Reviewer #2: No

4. Is the manuscript presented in an intelligible fashion and written in standard English?

Reviewer #1: Yes

Reviewer #2: Yes

5. Review Comments to the Author

Reviewer #1: This article investigates the mechanical response characteristics of Full Depth Reclaimed Pavement using the 3D-Move model, specifically focusing on pavements reclaimed with Portland cement (FDR-PC). The study constructs a numerical model to analyze mechanical behaviors under various conditions, including plane, spatial, speed, and temperature. The contribution of this research lies in providing a theoretical basis for the structural design of reclaimed pavements and offering insights for selecting specific technical solutions. It provides insights into the structural design and mechanical behavior of reclaimed pavements, which can inform practical engineering applications. However, minor improvements in language and clarity are needed:

1. Standardization of Units: The notation of units should be standardized—"Mpa" should be corrected to "MPa," and "KN/m³" should be unified as "kN/m³" in accordance with SI conventions.

2. Consistency in Author Names: The spelling of author names in the bibliography should be verified for consistency with those in the article descriptions.

3. Curve Labeling: The main curves in Figures 2 and 3 should be explicitly labeled with the corresponding reference temperatures for clarity.

4. Terminology Standardization: The term “grass-roots level” should be replaced with “recycled base layer” to maintain consistency in technical terminology.

5. Modeling Assumptions: The study assumes an infinite half-space structure for the residual layer of the old road, which may not fully align with real-world conditions. Given the complexity of actual boundary conditions, an assessment of how this assumption influences the study’s findings is necessary.

6. Time-History Analysis: In addition to describing the trends in stress-strain variations across structural layers based on curve shapes, a deeper exploration of the underlying mechanisms driving these trends should be provided.

7. Spatial Analysis Methodology: The spatial distribution of mechanical responses is analyzed by extracting eigenvalues from time-history variation curves. Clarification is needed regarding the rationale and selection criteria for these eigenvalues.

8. Climatic Considerations: The potential impact of regional climatic variations on long-term pavement performance should be evaluated when analyzing temperature effects.

9. Future Research Directions: Given the study's limitations, a discussion on future research directions should be included, outlining potential areas for further investigation.

Reviewer #2: This paper investigates the dynamic response of full-depth claimed asphalt pavement using 3D-move. The topic is import to for promoting the use of full-depth reclaimed asphalt. So comments regarding the details of the manuscript are given as follows:

1. Incorrect use of punctuation between keywords.

2. The order between ‘et al’ in the introduction and the number of the citation is not standardised.

3. Are there any limitations of the finite layer method in the 3D-Move programme when dealing with complex pavement structures (e.g. where there is an inhomogeneous material distribution)?

4. Chapter 4 section headings should be capitalised and the hierarchy confused.

5. In the dynamic analysis, is it sufficient to set the default travelling speed to 60km/h for the study? Should more speed conditions be added to the analysis because of the wide range of speed variations in different car models and actual road conditions?

6. In the conclusion,“within a certain range of speed” does not specify the range. Add a specific range of speed.

7. Numerical simulation has also been used to study the mechanical response of pavements, what are the specific aspects of this paper that are innovative in terms of research content and methodology?

6. PLOS authors have the option to publish the peer review history of their article (what does this mean? ). If published, this will include your full peer review and any attached files.

**Do you want your identity to be public for this peer review?** For information about this choice, including consent withdrawal, please see our Privacy Policy .

Reviewer #1: **Yes: ** Xiaogang Guo

Reviewer #2: No

---

## [Author Response · Author response to Decision Letter 0]

14 Mar 2025

Response to Reviewer 1 Comments

Dear Reviewer #1,

We would like to express our sincere appreciation for your helpful comments to improve this paper. We have made corresponding revision according to your advice. Revised portion are marked in red in the paper. The following are answers and revisions in response to your suggestions.

Comment 1:

Standardization of Units: The notation of units should be standardized—"Mpa" should be corrected to "MPa," and "KN/m³" should be unified as "kN/m³" in accordance with SI conventions.

Response 1:

Thank you for your thorough review and valuable feedback on the standardization of units in our manuscript. In response to your comments, we have carefully revised the manuscript in strict accordance with SI conventions:

All instances of “Mpa” have been corrected to “MPa”, with the prefix “M” capitalized.All occurrences of “KN/m³” have been unified as “kN/m³”, with “k” in lowercase and “N” capitalized.

We have systematically checked and updated these notations throughout the manuscript to ensure consistency. Your attention to detail is greatly appreciated.

Comment 2:

Consistency in Author Names: The spelling of author names in the bibliography should be verified for consistency with those in the article descriptions.

Response 2:

Thank you for highlighting the need for consistency in author name spellings between the article text and the bibliography. In response to your feedback, we have meticulously cross-verified all author names throughout the manuscript and reference list. Key revisions include:

Aligning the spelling, capitalization, and abbreviation of author names in both the main text and references;Correcting minor discrepancies caused by formatting errors or manual input oversights.

Comment 3:

Curve Labeling: The main curves in Figures 2 and 3 should be explicitly labeled with the corresponding reference temperatures for clarity.

Response 3:

Thank you for your valuable feedback on improving the clarity of curve labeling in Figures 2 and 3. We have carefully revised the figures as follows:

The corresponding reference temperature (20℃) is directly marked beside the curves in Figures 2 and 3 to ensure that readers can intuitively associate the curves with the temperature parameters.

Comment 4:

Terminology Standardization: The term “grass-roots level” should be replaced with “recycled base layer” to maintain consistency in technical terminology.

Response 4:

Thank you for emphasizing the need for standardized terminology. We have carefully addressed your comment as follows:

All instances of “grass-roots level” have been replaced with “recycled base layer” to align with established technical terminology in the field.

Comment 5:

Modeling Assumptions: The study assumes an infinite half-space structure for the residual layer of the old road, which may not fully align with real-world conditions. Given the complexity of actual boundary conditions, an assessment of how this assumption influences the study’s findings is necessary.

Response 5:

Thank you for your insightful comments on the applicability of the model assumptions. Assuming the residual layer of the old road as an infinite half-space structure does indeed differ from the actual situation. In reality, the boundary conditions of the residual layer of the old road are more complex, and there may be interactions with the surrounding soil and other situations. However, in this study, this assumption simplifies the establishment and calculation process of the model, and at the initial research stage, it helps to highlight the influence of the main factors on the mechanical response of the pavement.

Comment 6:

Time-History Analysis: In addition to describing the trends in stress-strain variations across structural layers based on curve shapes, a deeper exploration of the underlying mechanisms driving these trends should be provided.

Response 6:

Thank you very much for your valuable comment regarding the Time-History Analysis. We agree that a deeper exploration of the underlying mechanisms driving the trends in stress-strain variations across structural layers is essential. In response to your suggestion, we have summarized and revised section 4.1 of the manuscript.

Comment 7:

Spatial Analysis Methodology: The spatial distribution of mechanical responses is analyzed by extracting eigenvalues from time-history variation curves. Clarification is needed regarding the rationale and selection criteria for these eigenvalues.

Response 7:

Thank you for your insightful comment regarding the Spatial Analysis Methodology. We fully acknowledge the importance of clarifying the rationale and selection criteria for the eigenvalues used in analyzing the spatial distribution of mechanical responses.

In response to your suggestion, we have added the following content in our manuscript:

“Taking Fig. 11 as an example, for the case of tensile stress, the eigenvalue 1 can clarify the maximum degree of tensile at the point under wheel load, this is because tensile failure is one of the common failure modes in pavement structure design. By determining the eigenvalue of the maximum tensile degree, the ultimate tensile capacity of the material under wheel load can be evaluated; for the case of both compressive and tensile time course, the eigenvalue A and eigenvalue B indicate the peak value of positive and negative areas respectively. They are chosen as eigenvalue because in the actual force application process on the road surface, the material not only experiences tensile force but also compressive force, and the magnitude and variation of both forces have a significant impact on the fatigue life and overall performance of the road surface. These two eigenvalues can clearly reflect the extreme conditions of the material under different stress states, which is helpful for comprehensively evaluating the performance of the road surface under complex force conditions.”

Comment 8:

Climatic Considerations: The potential impact of regional climatic variations on long-term pavement performance should be evaluated when analyzing temperature effects.

Response 8:

Thank you for your astute comment regarding the Climatic Considerations. It is indeed crucial to evaluate the potential impact of regional climatic variations on the long-term pavement performance when analyzing temperature effects, as climate can exert a significant influence on pavement behavior over an extended period.

However, in our current study, the primary focus was placed on establishing the mechanical response model under typical conditions. Our intention was to first lay a solid foundation by understanding the fundamental mechanical responses in a controlled and representative scenario, which would serve as a basis for further investigations in the future.

Comment 9:

Future Research Directions: Given the study's limitations, a discussion on future research directions should be included, outlining potential areas for further investigation.

Response 9:

Thank you for your valuable suggestion regarding the Future Research Directions. We wholeheartedly agree that it is essential to discuss potential areas for further investigation given the limitations of our current study.

In response to your comment, we have added the following discussion on future research directions in our manuscript:

“Looking forward to the future, we can further deepen our understanding of recycled pavement from the following research directions. It is a potential research field to explore the influence of the mix proportion and construction technology of different recycled material mixtures on the mechanical properties of pavement. With the increasing emphasis on sustainable infrastructure construction, the role of full depth recycled pavement has attracted more and more attention. By studying this, it will contribute to the wide adoption and better performance of environmentally friendly pavement solutions.”

Response to Reviewer 2 Comments

Dear Reviewer #2,

We would like to express our sincere appreciation for your helpful comments to improve this paper. We have made corresponding revision according to your advice. Revised portion are marked in red in the paper. The following are answers and revisions in response to your suggestions.

Comment 1:

Incorrect use of punctuation between keywords.

Response 1:

Thank you for pointing out the issue regarding the incorrect use of punctuation between keywords. We have carefully reviewed and corrected these punctuation errors throughout the text to ensure proper grammar and readability.

Comment 2:

The order between ‘et al’ in the introduction and the number of the citation is not standardised.

Response 2:

Thank you for highlighting the issue regarding the non-standardised order between 'et al' and the citation number in the introduction. We have gone through the relevant parts and adjusted them to follow the correct standardised format.

Comment 3:

Are there any limitations of the finite layer method in the 3D-Move programme when dealing with complex pavement structures (e.g. where there is an inhomogeneous material distribution)?

Response 3:

Thank you for your incisive comment regarding the limitations of the finite layer method within the 3D-Move programme when handling complex pavement structures.

Indeed, the finite layer method in the 3D-Move programme has certain limitations when dealing with complex pavement structures characterized by inhomogeneous material distribution. However, in our current study, the pavement structure under consideration is relatively typical, mainly focusing on the mechanical responses of full-depth recycled pavements under common working conditions. In such scenarios where the pavement structure is layered and relatively homogeneous and our interest lies in the responses at specific positions, the finite layer method demonstrates a significant advantage in terms of computational efficiency.In view of the limitations of the model, the material parameters are simplified and assumed in order to meet the applicable conditions of the finite layer method.

Comment 4:

Chapter 4 section headings should be capitalised and the hierarchy confused.

Response 4:

Thank you for pointing out the issues regarding the capitalisation and hierarchy confusion of Chapter 4 section headings. We have carefully reviewed and made the necessary adjustments to capitalise the headings properly and clarify the hierarchical structure as per the required standards.

Comment 5:

In the dynamic analysis, is it sufficient to set the default travelling speed to 60km/h for the study? Should more speed conditions be added to the analysis because of the wide range of speed variations in different car models and actual road conditions?

Response 5:

Thank you for your astute comment regarding the setting of the travelling speed in the dynamic analysis.

In our study, the decision to set the default travelling speed at 60km/h was based on multiple rationales. Firstly, this speed aligns with the common design speeds of numerous typical roads. It represents a standard velocity that is frequently adopted in road engineering designs and is thus a reasonable benchmark for our analysis focused on understanding the general mechanical responses of the pavement under normal traffic conditions.

Secondly, it draws on the experience from a wealth of previous similar studies. Many prior investigations have utilized this or comparable speeds as a reference point to establish foundational knowledge about pavement mechanics under dynamic loads, and their findings have provided valuable references for our current research.

While it is undeniable that in real-world scenarios, different car models and actual road conditions lead to a wide range of speed variations, our extensive preliminary studies have shown that within a certain range of speeds, the influence trends of vehicle speed on the pavement's mechanical responses exhibit remarkable similarities. That is to say, although the specific magnitudes of the responses may vary with different speeds, the overall patterns and directions of change remain consistent. Therefore, choosing 60km/h as the default speed enables us to capture these general trends effectively and still gain valuable insights into the fundamental mechanical behavior of the pavement under dynamic loads.

Comment 6:

In the conclusion,“within a certain range of speed” does not specify the range. Add a specific range of speed.

Response 6:

Thank you for your comment regarding the specification of the speed range in the conclusion. We have taken your advice seriously and have now added “within the speed range of 20 km/h to 80 km/h” in the manuscript as you suggested.

Comment 7:

Numerical simulation has also been used to study the mechanical response of pavements, what are the specific aspects of this paper that are innovative in terms of research content and methodology?

Response 7:

Thank you for your insightful comment regarding the innovation aspects of our paper.

Our study indeed presents several innovative elements in both research content and methodology.

In terms of the research content, we specifically focus on the Full-Depth Recycled Pavement (FDR-PC). Currently, there is relatively limited research on the mechanical properties of this particular type of pavement. Our study fills this gap by providing crucial theoretical bases for its structural design and the selection of technical solutions. This enables a better understanding of how FDR-PC performs mechanically and offers valuable guidance for its practical application in the field of pavement engineering.

When it comes to the research methodology, we utilize the 3D-Move program to conduct a dynamic analysis of the pavement's mechanical response. What sets our approach apart is that we comprehensively consider four aspects, namely plane, space, speed, and temperature. Unlike previous studies that often focused on only a single factor or a limited number of factors, our multi-factor comprehensive analysis method is more holistic. By analyzing the characteristics of mechanical responses under different factors, we obtain richer and more comprehensive research results, which allows for a deeper exploration into the mechanical behavior of the Full-Depth Recycled Pavement.

We believe these innovative aspects contribute to the uniqueness and value of our study in the existing body of pavement research.

---

## [Decision Letter · Decision Letter 1]

3 Apr 2025

Mechanical response characteristics of Full Depth Reclaimed Pavement based on 3D-Move model

PONE-D-25-09395R1

Dear Dr. Zhang,

We’re pleased to inform you that your manuscript has been judged scientifically suitable for publication and will be formally accepted for publication once it meets all outstanding technical requirements.

Kind regards,

Jiaolong Ren

Academic Editor

PLOS ONE

Additional Editor Comments (optional):

Reviewers' comments:

Reviewer's Responses to Questions

**Comments to the Author**

1. If the authors have adequately addressed your comments raised in a previous round of review and you feel that this manuscript is now acceptable for publication, you may indicate that here to bypass the “Comments to the Author” section, enter your conflict of interest statement in the “Confidential to Editor” section, and submit your "Accept" recommendation.

Reviewer #1: All comments have been addressed

2. Is the manuscript technically sound, and do the data support the conclusions?

Reviewer #1: Yes

3. Has the statistical analysis been performed appropriately and rigorously? 

Reviewer #1: Yes

4. Have the authors made all data underlying the findings in their manuscript fully available?

Reviewer #1: Yes

5. Is the manuscript presented in an intelligible fashion and written in standard English?

Reviewer #1: Yes

6. Review Comments to the Author

Reviewer #1: All of the reviewers' concerns have been addressed comprehensively in this revision. The authors have meticulously gone through each comment and provided detailed responses, ensuring that the manuscript now meets the expected standards of clarity, accuracy, and scientific rigor.

7. PLOS authors have the option to publish the peer review history of their article (what does this mean? ). If published, this will include your full peer review and any attached files.

**Do you want your identity to be public for this peer review?** For information about this choice, including consent withdrawal, please see our Privacy Policy .

Reviewer #1: **Yes: ** Xiaogang Guo

---

## [Editor Report · Acceptance letter]

PONE-D-25-09395R1

PLOS ONE

Dear Dr. Zhang,

I'm pleased to inform you that your manuscript has been deemed suitable for publication in PLOS ONE. Congratulations! Your manuscript is now being handed over to our production team.

Kind regards,

on behalf of

Dr. Jiaolong Ren

Academic Editor

PLOS ONE